# DiffAug: A Diffuse-and-Denoise Augmentation for Training Robust Classifiers

**Chandramouli S. Sastry, Sri Harsha Dumpala, Sageev Oore**
Dalhousie University, Canada.

## Abstract

We introduce DiffAug, a simple and efficient diffusion-based augmentation technique to train image classifiers for the crucial yet challenging goal of improved classifier robustness. Applying DiffAug to a given example consists of one forward-diffusion step followed by one reverse-diffusion step. Using both ResNet-50 and Vision Transformer architectures, we comprehensively evaluate classifiers trained with DiffAug and demonstrate the surprising effectiveness of single-step reverse diffusion in improving robustness to covariate shifts, certified adversarial accuracy and out of distribution detection. When we combine DiffAug with other augmentations such as AugMix and DeepAugment we demonstrate further improved robustness. Finally, building on this approach, we also improve classifier-guided diffusion wherein we observe improvements in: (i) classifier-generalization, (ii) gradient quality (i.e., improved perceptual alignment) and (iii) image generation performance. We thus introduce a computationally efficient technique for training with improved robustness that does not require any additional data, and effectively complements existing augmentation approaches.

## 1 Introduction

Motivated by the success of diffusion models in high-fidelity and photorealistic image generation, generative data augmentation is an emerging application of diffusion models. While attempts to train improved classifiers with synthetic data have proved challenging, Azizi et al. [2] impressively demonstrated that extending the training dataset with synthetic images generated using Imagen [42] — with appropriate sampling parameters (e.g. prompt and guidance strength) — could indeed improve Imagenet classification. In a similar experiment with Stable Diffusion (SD) [41], Sariyildiz et al. [45] studied classifiers trained exclusively on synthetic images (i.e. no real images) and discovered improvements when training on a subset of 100 Imagenet classes. The success of generative data augmentation depends crucially on sample quality [40], so these findings irrefutably highlight the superior generative abilities of diffusion models.

Despite these impressive findings, widespread adoption of diffusion models for synthetic data augmentation is constrained by high computational cost of diffusion sampling, which requires multiple steps of reverse diffusion to ensure sufficient sample quality. Furthermore, both SD and Imagen are trained on upstream datasets much larger than Imagenet and some of these improvements could also be attributed to the quality and scale of the upstream dataset [19]. For example, Bansal and Grover [4] find limited advantages in synthetic examples generated from a diffusion model trained solely on Imagenet.

Together, these limitations motivate us to explore a diffusion-based augmentation technique that is not only computationally efficient but can also enhance classifier training without relying on extra data. To that end, we consider the following questions:

1. *Can we leverage a diffusion model trained with no extra data?*

38th Conference on Neural Information Processing Systems (NeurIPS 2024).

2. *Can we train improved classifiers with a single step of reverse diffusion?*

In the context of reverse diffusion sampling, the intermediate output obtained after one reverse diffusion (i.e., denoising) step is commonly interpreted as an *approximation* of the final image by previous works and has been utilised to define the guidance function at each step of guided reverse-diffusion (e.g., [3, 10, 11]). Similarly, Diffusion Denoised Smoothing (DDS) [5] applies denoised smoothing [43], a certified adversarial defense for pretrained classifiers, using one reverse diffusion step. In contrast to previous work, we use the output from a single reverse diffusion step as an augmentation to *train* classifiers (i.e., not just at inference time) as we describe next.

**Diffuse-and-Denoise Augmentation** Considering a diffusion model defined such that time $t = 0$ refers to the data distribution and time $t = T$ refers to isotropic Gaussian noise, we propose to generate augmentations of train examples by first applying a Gaussian perturbation (i.e., forward-diffusion to a random time $t \in [0, T]$) and then crucially, applying a single diffusion denoising step (i.e., one-step reverse diffusion). That is, we treat these diffused-and-denoised examples as augmentations of the original train image and refer to this technique as DiffAug. A one-step diffusion denoised example derived from a Gaussian perturbed train example can also be interpreted as an intermediate sample in *some* reverse diffusion sequence that starts with pure noise and ends at the train example. Interpreted this way, our classifier can be viewed as having been trained on partially-synthesized images whose ostensible quality varies from unrecognizable (DiffAug using $t \approx T$) to excellent (DiffAug using $t \approx 0$). This is surprising because, while Ravuri and Vinyals [40] find that expanding the train dataset even with a small fraction of (lower quality) synthetic examples can lead to noticeable drops in classification accuracy, we find that classifier accuracy over test examples does not degrade despite being explicitly trained with partially synthesized train images. Instead, we show that diffusion-denoised examples offer a regularization effect when training classifiers that leads to *improved classifier robustness without sacrificing clean test accuracy and without requiring additional data*.

Our contributions in this work are as follows[1]:

(a) **DiffAug** We propose DiffAug, a simple, efficient and effective diffusion-based augmentation technique. We provide a qualitative and analytical discussion on the unique regularization effect — complementary to other leading and classic augmentation methods — introduced by DiffAug.

(b) **Robust Classification**. Using both ResNet-50 and ViT architectures, we evaluate the models in terms of their robustness to covariate shifts, adversarial examples (i.e., certified accuracy under Diffusion Denoised Smoothing (DDS)[5]) and out-of-distribution detection.

(c) **DiffAug-Ensemble (DE)** We extend DiffAug to test-time and introduce DE, a simple test-time image augmentation/adaptation technique to improve robustness to covariate shift that is not only competitive with DDA [18], the state-of-the-art image adaptation method but also 10x faster.

(d) **Perceptual Gradient Alignment.** Motivated by the success of DDS and evidence of perceptually aligned gradients (PAGs) in robust classifiers, we qualitatively analyse the classifier gradients and discover the perceptual alignment described in previous works. We then theoretically analyse the gradients through the score function to explain this perceptual alignment.

(e) **Improved Classifier-Guided Diffusion**. Finally, we build on (d) to improve gradient quality in guidance classifiers and demonstrate improvements in terms of: (i) generalization, (ii) perceptual gradient alignment and (iii) image generation performance.

## 2 Background

The stochastic diffusion framework [47] consists of two key components: 1) the forward-diffusion (i.e., data to noise) stochastic process, and 2) a learnable score-function that can then be used for the reverse-diffusion (i.e., noise to data) stochastic process.

The forward diffusion stochastic process $\{\mathbf{x}_t\}_{t \in [0,T]}$ starts at data, $\mathbf{x}_0$, and ends at noise, $\mathbf{x}_T$. We let $p_t(\mathbf{x})$ denote the probability density of $\mathbf{x}$ at time $t$ such that $p_0(\mathbf{x})$ is the data distribution, and $p_T(\mathbf{x})$ denotes the noise distribution. The diffusion is defined with a stochastic-differential-equation (SDE):

$$d\mathbf{x} = \mathbf{f}(\mathbf{x}, t) \, dt + g(t) \, d\mathbf{w}, \qquad (1)$$

where $\mathbf{w}$ denotes a standard Wiener process, $\mathbf{f}(\mathbf{x}, t)$ is a drift coefficient, and $g(t)$ is a diffusion coefficient. The drift and diffusion coefficients are usually specified manually such that the solution to

---

[1]Code available at `https://github.com/oore-lab/diffaug`

the SDE with initial value $\mathbf{x}_0$ is a time-varying Gaussian distribution $p_t(\mathbf{x}|\mathbf{x}_0)$ whose mean $\mu(\mathbf{x}_0, t)$ and standard deviation $\sigma(t)$ can be exactly computed.

To sample from $p_0(\mathbf{x})$ starting with samples from $p_T(\mathbf{x})$, we solve the reverse diffusion SDE [1]:

$$d\mathbf{x} = [\mathbf{f}(\mathbf{x}, t) - g(t)^2 \nabla_\mathbf{x} \log p_t(\mathbf{x})] \, dt + g(t) \, d\bar{\mathbf{w}}, \tag{2}$$

where $d\bar{\mathbf{w}}$ is a standard Wiener process when time flows from T to 0, and $dt$ is an infinitesimal negative timestep. In practice, the score function $\nabla_\mathbf{x} \log p_t(\mathbf{x})$ is estimated by a neural network $s_\theta(\mathbf{x}, t)$, parameterized by $\theta$, trained using a score-matching loss [47].

**Denoised Examples.** Given $(\mathbf{x}_0, y) \sim p_0$ and $\mathbf{x} \sim p_t(\mathbf{x}|\mathbf{x}_0) = \mathcal{N}(\mathbf{x} \mid \mu(\mathbf{x}_0, t), \ \sigma^2(t)\mathbf{I})$, we can compute the denoised image $\hat{\mathbf{x}}_t$ using the pretrained score network $s_\theta$ as:

$$\hat{\mathbf{x}}_t = \mathbf{x} + \sigma^2(t) s_\theta(\mathbf{x}, t) \tag{3}$$

Intuitively, $\hat{\mathbf{x}}_t$ is an *expectation* over all possible images $\mathbf{m}_t = \mu(\mathbf{x}_0, t)$ that are *likely* to have been perturbed with $\mathcal{N}(\mathbf{0}, \ \sigma^2(t)\mathbf{I})$ to generate $\mathbf{x}$ and the denoised example $\hat{\mathbf{x}}_t$ can be written as

$$\hat{\mathbf{x}}_t = \mathbb{E}[\mathbf{m}_t|\mathbf{x}] = \int_{\mathbf{m}_t} \mathbf{m}_t \, p_t(\mathbf{m}_t|\mathbf{x}) d\mathbf{m}_t \tag{4}$$

We note that the mean does not change with diffusion time $t$ in variance-exploding SDEs while the mean decays to zero with diffusion time for variance-preserving SDEs (DDPMs).

## 3 DiffAug: Diffuse-and-Denoise Augmentation

In this section, we describe Diffuse-and-Denoise Augmentation (DiffAug, in short) and then provide an analytical and qualitative discussion on the role of denoised examples in training classifiers. While we are not aware of any previous study on training classifiers using partially denoised examples, Diffusion-denoised smoothing (DDS) [5], DiffPure [36] and Diffusion Driven Adaptation (DDA) [18] are test-time applications of — single-step (DDS) and multi-step (DiffPure/DDA) — denoised examples to promote robustness in pretrained classifiers.

As implied by its name, DiffAug consists of two key steps: (i) Diffuse: first, we diffuse a train example $\mathbf{x}_0$ to a uniformly sampled time $t \sim \mathcal{U}(0, T)$ and generate $\mathbf{x} \sim p_t(\mathbf{x}|\mathbf{x}_0)$; (ii) Denoise: then, we denoise $\mathbf{x}$ using a single application of trained score network $s_\theta$ as shown in Eq. 3 to generate $\hat{\mathbf{x}}_t$. We assume that the class label does not change upon augmentation (see discussion below) and train the classifier to minimize the following cross-entropy loss:

$$\mathcal{L} = \mathbb{E}_{t,\mathbf{x}_0}[-\log p_\phi(y|\hat{\mathbf{x}}_t)] \tag{5}$$

$\mathbf{x}_0 \longrightarrow \qquad\qquad \hat{\mathbf{x}}_t \longrightarrow$

Figure 1: DiffAug Augmentations. The leftmost column shows four original training examples ($\mathbf{x}_0$); to the right of that, we display 8 random augmentations ($\hat{\mathbf{x}}_t$) for each image between $t = 350$ and $t = 700$ in steps of size 50. Augmentations generated for $t < 350$ are *closer* to the input image while the augmentations for $t > 700$ are *farther* from the input image. We observe that the diffusion denoised augmentations with larger values of $t$ do not preserve the class label introducing noise in the training procedure. However, we find that this does not lead to empirical degradation of classification accuracy but instead contributes to improved robustness. Also, see Fig. 6 in appendix for a toy 2d example.

where, $t \sim \mathcal{U}(0, T)$, $(\mathbf{x}_0, y) \sim p_0(\mathbf{x})$, and $p_\phi$ denotes the classifier parameterized by $\phi$. In this work, we show the effectiveness of DiffAug as a standalone augmentation technique, as well as the further compounding effect of combining it with robustness-enhancing techniques such as Augmix and DeepAugment, showing that DiffAug is achieving a robustness not captured by the other approaches. When combining DiffAug with such novel augmentation techniques, we simply include Eq. 5 as an additional optimization objective instead of stacking augmentations (for example, we can alternatively apply DiffAug to images augmented with Augmix/DeepAugment or vice-versa). Also, our preliminary analysis on stacking augmentations showed limited gains over simply training the network to classify independently augmented samples likely because training on independent augmentations implicitly generalizes to stacked augmentations.

**Qualitative Analysis and Manifold Theory.** When generating augmentations, it is important to ensure that the resulting augmentations lie on the image manifold. Recent studies [10, 38] on theoretical properties of denoised examples suggest that denoised examples can be considered to

be on the data manifold under certain assumptions lending theoretical support to the idea of using denoised examples as augmentations. We can interpret training on denoised examples as a type of Vicinal Risk Minimization (VRM) since the denoised examples can be considered to lie in the *vicinal distribution* of training samples. Previous works have shown that VRM improves generalization: for example, Chapelle et al. [8] use Gaussian perturbed examples ($\mathbf{x}$) as the vicinal distribution while MixUp [57] uses a convex sum of two random inputs (and their labels) as the vicinal distribution. From Eq. 4, we can observe that a denoised example is a convex sum over $\mathbf{m}_t$ and we can interpret $\hat{\mathbf{x}}_t$ as being *vicinal* to examples $\mathbf{m}_t$ that have a non-trivial likelihood, $p_t(\mathbf{m}_t|\mathbf{x})$, of generating $\mathbf{x}$. The distribution $p_t(\mathbf{m}_t|\mathbf{x})$ is concentrated around examples perceptually similar to $\mu(\mathbf{x}_0, t)$ when $\mathbf{x}$ is closer to $\mathbf{x}_0$ (i.e., smaller $\sigma(t)$) and becomes more entropic as the noise scale increases: we can qualitatively observe this in Fig. 1.

Diffusion denoised augmentations generated from larger $\sigma(t)$ can introduce label-noise into the training since the class-labels may not be preserved upon augmentation – for example, some of the diffusion denoised augmentations of the dog in Fig. 1 resemble architectural buildings. Augmentations that alter the true class-label are said to cause manifold intrusion [21] leading to underfitting and lower classification accuracies. In particular, accurate predictions on class-altered augmented examples would be incorrectly penalised causing the classifier to output less confident predictions on all inputs (i.e., underfitting). Interestingly, however, diffusion denoised augmentations that alter the true-class label are also of lower sample quality. The correlation between label noise and sample quality allows the model to selectively lower its prediction confidence when classifying denoised samples generated from larger perturbations applied to $\mathbf{x}_0$ (we empirically confirm this in Section 4). In other words, the classifier *learns* to observe important details in $\hat{\mathbf{x}}_t$ to determine the optimal prediction estimating the class-membership probabilities of $\mathbf{x}_0$. On the other hand, any augmentation that alters class-label by preserving the sample quality can impede the classifier training since the classifier cannot rely on visual cues to selectively lower its confidence (for an example, see Fig. 7 in appendix). Therefore, we do not consider multi-step denoising techniques to generate augmentations — despite their potential to improve sample quality — since this would effectively decorrelate label-noise and sample-quality necessitating additional safeguards — e.g., we would then need to determine the maximum diffusion time we could use for augmentation without altering the class-label or scale down the loss terms corresponding to samples generated from larger $\sigma(t)$. We leave this exploration to future work and include a preliminary analysis in Appendix B.2.2.

**Test-time Augmentation with DiffAug.** Test-time Augmentation (TTA) [30] is a technique to improve classifier prediction using several augmented copies of a single test example. A simple yet successful TTA technique is to just average the model predictions for each augmentation of a test sample. We extend DiffAug to generate test-time augmentations of a test-example wherein we apply DiffAug using different values of diffusion times $t$ and utilize the average predictions across all the augmentations to classify the test example $\mathbf{x}_0$:

$$p(y|\mathbf{x}_0) = \frac{1}{|\mathcal{S}|} \sum_{t \in \mathcal{S}} p_\phi(y|\hat{\mathbf{x}}_t) \tag{6}$$

where, $\mathcal{S}$ denotes the set of diffusion times considered. We refer to this as DiffAug Ensemble (DE). A forward diffusion step followed by diffusion denoising can be interpreted as projecting a test-example with unknown distribution shift into the source distribution and forms the basis of DDA, a diffusion-based image adaptation technique. Different from DE, DDA uses a novel multi-step denoising technique to transform the diffused test example into the source distribution. Since DE uses single-step denoised examples of forward diffused samples, we observe significant improvement in terms of running time while either improving over or remaining on par with DDA.

## 4 Experiments I: Classifier Robustness

In this section, we evaluate classifiers trained with DiffAug in terms of their standard classification accuracy as well as their robustness to distribution shifts and adversarial examples. We primarily conduct our experiments on Imagenet-1k and use the unconditional $256 \times 256$ Improved-DDPM [14, 35] diffusion model to generate the augmentations. We apply DiffAug to train the popular ResNet-50 (RN-50) backbone as well as the recent Vision-Transformer (ViT) model (ViT-B-16, in particular). In addition to extending the default augmentations used to train RN-50/ViT with DiffAug, we also combine our method with the following effective robustness-enhancing augmentations: (i) AugMix (AM), (ii) DeepAugment (DA) and (iii) DeepAugment+AugMix (DAM). While we train

Table 1: Top-1 Accuracy (%) on Imagenet-C (severity=5) and Imagenet-Test. We summarize the results for each combination of Train-augmentations and evaluation modes. The average (avg) accuracies for each classifier and evaluation mode is shown.

| Train Augmentations | ImageNet-C (severity = 5) | | | | | ImageNet-Test | | | | |
|---|---|---|---|---|---|---|---|---|---|---|
| | DDA | DDA (SE) | DE | Def. | **Avg** | DDA | DDA (SE) | DE | Def. | **Avg** |
| AM | 33.18 | 36.54 | 34.08 | 26.72 | 32.63 | 62.23 | 75.98 | 73.8 | 77.53 | 72.39 |
| AM+DiffAug | 34.64 | 38.61 | 38.58 | 29.47 | **35.33** | 63.53 | 76.09 | 75.88 | 77.34 | **73.21** |
| DA | 35.41 | 39.06 | 37.08 | 31.93 | 35.87 | 63.63 | 75.39 | 74.28 | 76.65 | 72.49 |
| DA+DiffAug | 37.61 | 41.31 | 40.42 | 33.78 | **38.28** | 65.47 | 75.54 | 75.43 | 76.51 | **73.24** |
| DAM | 40.36 | 44.81 | 41.86 | 39.52 | 41.64 | 65.54 | 74.41 | 73.54 | 75.81 | 72.33 |
| DAM+DiffAug | 41.91 | 46.35 | 44.77 | 41.24 | **43.57** | 66.83 | 74.64 | 74.39 | 75.66 | **72.88** |
| RN50 | 28.35 | 30.62 | 27.12 | 17.87 | 25.99 | 58.09 | 74.38 | 71.43 | 76.15 | 70.01 |
| RN50+DiffAug | 31.15 | 33.51 | 32.22 | 20.87 | **29.44** | 61.04 | 74.87 | 75.07 | 75.95 | **71.73** |
| ViT-B/16 | 43.6 | 52.9 | 48.25 | 50.75 | 48.88 | 67.4 | 81.72 | 80.43 | 83.71 | 78.32 |
| ViT-B/16+DiffAug | 45.05 | 53.54 | 51.87 | 52.78 | **50.81** | 70.05 | 81.85 | 82.59 | 83.59 | **79.52** |
| **Avg** | 37.13 | 41.73 | 39.63 | 34.49 | 38.24 | 64.38 | 76.49 | 75.68 | 77.89 | 73.61 |
| **Avg (No-DiffAug)** | 36.18 | 40.79 | 37.68 | 33.36 | 37.00 | 63.38 | 76.38 | 74.70 | 77.97 | 73.11 |
| **Avg (DiffAug)** | 38.07 | 42.66 | 41.57 | 35.63 | 39.48 | 65.38 | 76.60 | 76.67 | 77.81 | 74.12 |

the RN-50 from scratch, we follow DeIT-III recipe[50] for training ViTs and apply DiffAug in the second training stage; when combining with AM/DA/DAM, we finetune the official checkpoint for 10 epochs. More details are included in Appendix B.1. In the following, we will evaluate the classifier robustness to **(i)** covariate shifts, **(ii)** adversarial examples and **(iii)** out-of-distribution examples.

**Covariate Shifts** To evaluate the classifiers trained with/without DiffAug in terms of their robustness to covariate-shifts, we consider the following evaluation modes:

(a) **DDA**: A diffusion-based test-time image-adaptation technique to transform the test image into the source distribution.
(b) **DDA-SE**: We consider the original test example as well as the DDA-adapted test-example by averaging the classifier predictions following the self-ensemble (SE) strategy proposed in [18].
(c) **DiffAug-Ensemble (DE)**: We use a set of test-time DiffAug augmentations to classify a test example as described in Eq. (6). Following DDA, we determine the following range of diffusion times $\mathcal{S} = \{0, 50, \ldots, 450\}$. In other words, we generate 9 DiffAug augmentations for each test example.
(d) **Default**: In the default mode, we directly evaluate the model on the test examples.

We evaluate the classifiers on Imagenet-C, a dataset of 15 synthetic corruptions applied to Imagenet-test and summarize the results across all evaluation modes in Table 9. We summarize our observations as follows:

 (i) **DiffAug introduces consistent improvements** Classifiers trained with DiffAug consistently improve over their counterparts trained without these augmentations across all evaluation modes. The average relative improvements across all corruptions range from 5.3% to 28.7% in the default evaluation mode (see Table 6 in Appendix). On clean examples, we observe that DiffAug helps minimize the gap between default evaluation mode and other evaluation modes while effectively preserving the default accuracy.
 (ii) **DE improves over DDA** On average, DiffAug Ensemble (DE) yields improved detection rate as compared to direct evaluation on DDA images. Furthermore, DiffAug-trained classifiers evaluated using DE improve on average over their counterparts (trained without DiffAug) evaluated using DDA-SE. This experiment interestingly reveals that a set of one-step diffusion denoised images (DE) can achieve improvements comparable to multi-step diffusion denoised images (DDA) at a substantially faster ($\sim 10$x) wallclock time (see Table 10 in Appendix).

**DiffAug vs Extra Synthetic Data**: Bansal and Grover [4] demonstrate that foundation models such as Stable-Diffusion can be used to generate additional synthetic data to improve classifier robustness to covariate shifts. We note that extra synthetic data with diffusion models trained exclusively on ImageNet do not help in enhancing classifier robustness (see Appendix B.2.1). In particular, Bansal and Grover [4] utilise diverse prompts to generate a synthetic clone of Imagenet consisting of 1.3M

Table 2: Top-1 Accuracy (%) across different types of distribution shifts when additional high-quality synthetic data from Stable-Diffusion is available (denoted by +Synth). We show the net improvement obtained by DiffAug training and DiffAug-Ensemble (DE) inference. For reference, we also include the results for the corresponding ResNet50 models without extra synthetic data.

| Model | ImageNet-C (Severity=5) | ImageNet-R | ImageNet-S | ImageNet Sketch | ImageNet-A | ImageNet-D | Average |
|---|---|---|---|---|---|---|---|
| RN50 | 17.87 | 36.16 | 7.12 | 24.09 | 0.03 | 11.36 | 16.10 |
| +DiffAug/DE | **32.22 (+14.85)** | **41.61 (+5.45)** | **12.52 (+5.40)** | **26.67 (+2.56)** | **1.09 (+1.06)** | 11.37 (+0.01) | **20.90** |
| RN50+Synth | 17.58 | 49.28 | 7.68 | 35.45 | 0.63 | 17.52 | 21.35 |
| +DiffAug/DE | **30.06 (+12.48)** | **54.71 (+5.43)** | **13.57 (+5.89)** | **37.39 (+1.94)** | **1.53 (+0.9)** | **21.41 (+3.89)** | **26.45** |

images. We utilize their open-sourced synthetic dataset and fine-tune the torchvision resnet-50 model for 10 epochs using both real and synthetic images and call this RN50+Synth. Next, we repeat this finetuning process with DiffAug and call this RN50+Synth+DiffAug. In Table 2, we summarize the results across many datasets and interestingly observe that DiffAug — using an ImageNet-only diffusion model — offers improvements over and beyond additional synthetic data using Stable-Diffusion. This is *surprising* due to the following reasons: **(i)** SD is trained on LAION-5B, a much larger dataset that also subsumes ImageNet. **(ii)** Additional synthetic data requires more compute per each sample (e.g., 50 reverse-diffusion steps) whereas DiffAug just uses one reverse-diffusion step.

While the improvements offered by the extra synthetic data can be largely attributed to the upstream training dataset, DiffAug augmentations offer complementary regularization benefits. To understand this, we first note that DiffAug augmentations are qualitatively distinct from high-quality synthetic images and can be interpreted as lying on the image manifold in regions between high-quality samples: depending on the diffusion time, the DiffAug augmentation can vary greatly in quality (as shown in Fig. 1). As a result, classifying some of these augmented images is more challenging as compared to the original examples producing a regularizing effect that leads to empirical robustness improvements.

We include a detailed evaluation across different datasets including ImageNet-R/S in the Appendix (Table 7). Overall, we observe that DiffAug training and DE inference help significantly enhance robustness to covariate-shifts in several cases.

**Out-of-Distribution (OOD) Detection.** Test examples whose labels do not overlap with the labels of the train distribution are referred to as out-of-distribution examples. To evaluate the classifiers in terms of their OOD-detection rates, we use the Imagenet near-OOD detection task defined in the OpenOOD benchmark, which also includes an implementation of recent OOD detection algorithms such as ASH[15], ReAct[49], Scale[54] and MSP[23]. For context, while the torchvision ResNet-50 checkpoint is most commonly used to evaluate new OOD detection algorithms, AugMix provides the best OOD detection amongst the existing robustness-enhancing augmentation techniques and AugMix/ASH is placed 3rd amongst 73 methods on the OpenOOD leaderboard (ordered by near-OOD performance). Yet in Table 3 we observe that DiffAug introduces *further* improvement on the challenging near-OOD detection task across all considered OOD algorithms.

Comparing our results to the leaderboard, we observe AugMix+DiffAug/Scale achieves an AUROC of 84.81 outperforming the second best method (84.01 AUROC) and comparable to the top AUROC of 84.87.

Table 3: AUROC on Imagenet Near-OOD Detection.

| Train Augmentation | ASH | MSP | ReAct | Scale | **Avg.** |
|---|---|---|---|---|---|
| AugMix(AM) | 82.16 | 77.49 | 79.94 | 83.61 | 80.8 |
| AM+DiffAug | 83.62 | 78.35 | 81.29 | 84.81 | **82.02** |
| RN50 | 78.17 | 76.02 | 77.38 | 81.36 | 78.23 |
| RN50+DiffAug | 79.86 | 76.86 | 78.76 | 82.81 | **79.57** |

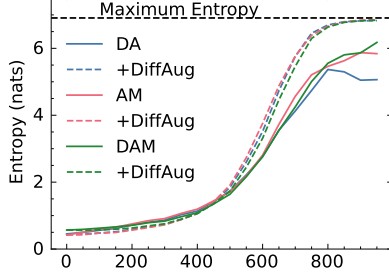

Figure 2: Average prediction entropy on DiffAug samples vs diffusion time measured with Imagenet-Test. We observe that the models trained with DiffAug correctly yield predictions with higher entropies (lower confidence) for images containing imperceptible details (i.e. larger $t$). Surprisingly, the classifiers trained without DiffAug do not also assign random-uniform label distribution for DiffAug images at $t = 999$, which have no class-information by construction. Also, see Fig. 11.

DiffAug training *teaches* the network to selectively lower its prediction confidence (higher prediction entropy) based on the image content (Fig. 2) and we hypothesize that this leads to improved OOD detection rates. We include the detailed OOD detection results in Appendix B.4. Interestingly, the combination of augmentations that improve robustness on covariate shifts may not necessarily lead to improved OOD detection rates: for example, AugMix+DeepAugment improves over both AugMix and DeepAugment on covariate shift but achieves lower OOD detection rates than either. On the other hand, we observe that combining with DiffAug enhances both OOD detection as well as robustness to covariate shift.

**Certified Adversarial Accuracy.** Denoised smoothing [43] is a certified defense for pretrained classifiers inspired from Randomized smoothing [12] wherein noisy copies of a test image are first denoised and then used as classifier input to estimate both the class-label and robust radius for each example. Using the same diffusion model as ours, DDS[5] already achieves state-of-the-art certified Imagenet accuracies with a pretrained 305M-parameter BeIT-L. Here, we evaluate the improvement in certified accuracy when applying DDS to a model trained with DiffAug and include the results in Appendix B.3. We speculate that finetuning the BeIT-L model with DiffAug should lead to similar improvements but skip this experiment since it is computationally expensive.

**Perceptually Aligned Gradients and Robustness.** Classifier gradients ($\nabla_{\mathbf{z}} \log p_\phi(y|\mathbf{z})$ where $\mathbf{z}$ is an image) which are semantically aligned with human perception are said to be perceptually aligned gradients (PAG) [17]. While input-gradients of a typical image classifier are usually unintelligible, gradients obtained from adversarially robust classifiers trained using randomized smoothing [28] or adversarial training [16, 44, 52] are perceptually-aligned. Motivated by the state-of-the-art certified adversarial accuracy achieved by DDS, we analyse the classifier gradients of one-step diffusion denoised examples — i.e., we analyse $\nabla_{\mathbf{x}} \log p_\phi(y|\hat{\mathbf{x}}_t)$ where $\mathbf{x} = p_t(\mathbf{x}|\mathbf{x}_0)$ and $\hat{\mathbf{x}}_t = \mathbf{x} + \sigma^2(t)s_\theta(\mathbf{x}, t)$. We visualise the gradients in Fig. 3 and interestingly discover the same perceptual alignment of gradients discussed in previous works (we compare with gradients of classifiers trained with randomized smoothing later). To theoretically analyse this effect, we first decompose the input-gradient using chain rule as:

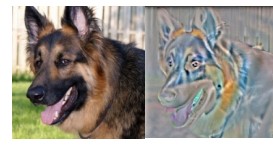

Figure 3: PAG example using ViT+DiffAug. We diffuse the Imagenet example (left) to $t = 300$ and visualise the min-max normalized classifier gradients (right). For easy viewing, we apply contrast maximization. More examples are shown below.

$$\frac{d \log p_\phi(y|\hat{\mathbf{x}}_t)}{d\mathbf{x}} = \frac{d \log p_\phi(y|\hat{\mathbf{x}}_t)}{d\hat{\mathbf{x}}_t} \frac{d\hat{\mathbf{x}}_t}{d\mathbf{x}} \tag{7}$$

Empirically, we find that the perceptual alignment is introduced due to transformation by $\frac{d\hat{\mathbf{x}}_t}{d\mathbf{x}}$ and analyse it further:

**Theorem 4.1.** *Consider a forward-diffusion SDE defined as in Eq. 1 such that $p_t(\mathbf{x}|\mathbf{x}_0) = \mathcal{N}(\mathbf{x} \mid \mathbf{m}_t, \sigma^2(t)I)$ where $\mathbf{m}_t = \mu(\mathbf{x}_0, t)$. If $\mathbf{x} \sim p_t(\mathbf{x})$ and $\hat{\mathbf{x}}_t = \mathbf{x} + \sigma^2(t)s_\theta(\mathbf{x}, t)$, for optimal parameters $\theta$, the derivative of $\hat{\mathbf{x}}_t$ w.r.t. $\mathbf{x}$ is proportional to the covariance matrix of the conditional distribution $p(\mathbf{m}_t|\mathbf{x})$. See proof in Appendix B.6.*

$$\frac{\partial \hat{\mathbf{x}}_t}{\partial \mathbf{x}} = J = \frac{1}{\sigma^2(t)} Cov[\mathbf{m}_t|\mathbf{x}]$$

This theorem shows us that the multiplication by $\frac{\partial \hat{\mathbf{x}}_t}{\partial \mathbf{x}}$ in Eq (7) is in fact a transformation by the covariance matrix $Cov[\mathbf{m}_t|\mathbf{x}]$. Multiplying a vector by $Cov[\mathbf{m}_t|\mathbf{x}]$ *stretches* the vector along the principal directions of the conditional distribution $p(\mathbf{m}_t|\mathbf{x})$. Intuitively, since the conditional distribution $p(\mathbf{m}_t|\mathbf{x})$ corresponds to the distribution of *candidate* denoised images, the principal directions of variation are perceptually aligned (to demonstrate, we apply SVD to $J$ and visualise the principal components in Appendix B.7) and hence stretching the gradient along these directions will yield perceptually aligned gradients. We note that our derivation complements Proposition 1 in Chung et al. [10] which proves certain properties (e.g., $J = J^\top$) of this derivative. In practice, however, the score-function is parameterized by unconstrained, flexible neural architectures that do not have exactly symmetric jacobian matrices $J$. For more details on techniques to enforce conservative properties of score-functions, we refer the reader to Chao et al. [7]. Ganz et al. [17] demonstrate that training a classifier to have perceptually aligned gradients also improves its robustness exposing the bidirectional relationship between robustness and PAGs. This works offers additional evidence supporting the co-occurrence of robustness and PAGs since we observe that classification of diffused-and-denoised images (e.g., DDS, DE, DDA) not only improve robustness but also produce PAGs.

**Ablation Analysis.** Appendix B.5 includes an ablation study on the following:

(a) **Extra Training**. The pretrained DA, AM, and DAM classifiers are sufficiently trained for 180 epochs and hence, we compare the DiffAug finetuned model directly with the pretrained checkpoint. For completeness, we train AugMix for another 10 epochs and confirm that there is no notable change in performance as compared to results in Tables 3, 7 and 9.

(b) **DiffAug Hyperparameters**. In our experiments, we considered the complete range of diffusion time. We investigate a simple variation where we either use $t \in [0, 500]$ or $t \in [500, 999]$ to generate the DiffAug augmentations.

(c) **DiffAug-Ensemble Hyperparameters**. We analyse how the choice of diffusion times considered in the set $S$ (Eq. (6)) affects DE performance.

(d) **Conditional DiffAug**. While our DiffAug experiments mainly utilize unconditional diffusion models, we can also utilize conditional diffusion models and explore an extension of AugMix with conditional DiffAug for various guidance strengths.

(e) **Latent-Space Diffusion Models**. We also tried the DiffAug augmentation method with Diffusion-Transformer (DiT): while we observed that the test-accuracy is preserved even in this case, we only observe slight robustness improvements as compared with pixel-space diffusion models. This may be explained by noting that the VAE-decoder is trained to output perceptually high-quality image and hence, does not accurately capture the expectation in Eq. (4).

## 5  Experiments II: Classifier-Guided Diffusion

Classifier guided (CG) diffusion is a conditional generation technique to generate class-conditional samples with an unconditional diffusion model. To achieve this, a time-conditional classifier is separately trained to classify noisy samples from the forward diffusion and we refer to this as a ***guidance classifier***

$$\mathcal{L}_{\text{CE}} = \mathbb{E}_{t,\mathbf{x}}[-\log p_\phi(y|\mathbf{x}, t)] \tag{8}$$

where, $t \sim \mathcal{U}(0, T)$, $\mathbf{x} \sim p_t(\mathbf{x}|\mathbf{x}_0)$ and $(\mathbf{x}_0, y) \sim p_0(\mathbf{x})$. At each step of the classifier-guided reverse diffusion (Eq. (2)), the guidance classifier is used to compute the class-conditional score $\nabla_{\mathbf{x}} \log p_t(\mathbf{x}|y) = \nabla_{\mathbf{x}} \log p_\phi(y|\mathbf{x}, t) + \lambda_s \nabla_{\mathbf{x}} \log p_t(\mathbf{x})$ ($\lambda_s$ is classifier scale[14]), which is used in place of unconditional score $\nabla_{\mathbf{x}} \log p_t(\mathbf{x})$.

**Denoising-Augmented (DA) Classifier**. The guidance classifiers participate in the sampling through their gradients, which indicate the pixel-wise perturbations that maximizes log-likelihood of the target class. Perceptually aligned gradients that resemble images from data distribution lead to meaningful pixel-wise perturbations that could potentially improve classifier-guidance and forms the motivation of Kawar et al. [29], where they propose an adversarial training recipe for guidance classifiers. With the same motivation, we instead build on Theorem 4.1 in order to improve perceptual alignment and propose to train guidance classifiers with denoised examples $\hat{\mathbf{x}}_t$ derived from $\mathbf{x}$. While the obvious choice is to simply train the guidance-classifier on $\hat{\mathbf{x}}_t$ instead of $\mathbf{x}$, we choose to provide both $\mathbf{x}$ as well as $\hat{\mathbf{x}}_t$ as simultaneous inputs to the classifier and instead optimize $\mathcal{L}_{\text{CE}} = \mathbb{E}_{t,\mathbf{x}}[-\log p_\phi(y|\mathbf{x}, \hat{\mathbf{x}}_t, t)]$ (compare with Eq. (8)). We preserve the noisy input since the primary goal of guidance classifiers is to classify noisy examples and this approach enables the model to flexibly utilize information from both inputs. We refer to guidance-classifiers trained using both $\mathbf{x}$ and $\hat{\mathbf{x}}_t$ as ***denoising-augmented (DA) classifier*** and use ***noisy classifiers*** to refer to guidance-classifiers trained exclusively on $\mathbf{x}$.

**Experiment setup.** We conduct our experiments on CIFAR10 and Imagenet and evaluate the advantages of DA-Classifiers over noisy classifiers. While we use the same Imagenet diffusion model described in Section 4, we use the deep NCSN++ (continuous) model released by Song et al. [47] as the score-network for CIFAR10 (VE-Diffusion). As compared to the noisy classifier, the DA-classifier has an additional input-convolution layer to process the denoised input and is identical from the second layer onwards. We describe our classifier architectures and the training details in Appendix C.1.

**Classification Accuracy.** We first compare guidance classifiers in terms of test accuracies as a measure of their generalization (Table 4) and find that DA-classifiers generalize better to the test data. Training classifiers with Gaussian perturbed examples often leads to underfitting [59] explaining the lower test accuracy observed with noisy classifiers. Interestingly, the additional denoised example helps address the underfitting – for example, see Fig. 19 (in appendix). One explanation of this finding could be found in Chung et al. [10],

Table 4: Summary of Test Accuracies for CIFAR10 and Imagenet: each test example is diffused to a random uniformly sampled diffusion time. Both classifiers are shown the same diffused example.

| Method | CIFAR10 | Imagenet |
|---|---|---|
| Noisy Classifier | 54.79 | 33.78 |
| DA-Classifier | **57.16** | **36.11** |

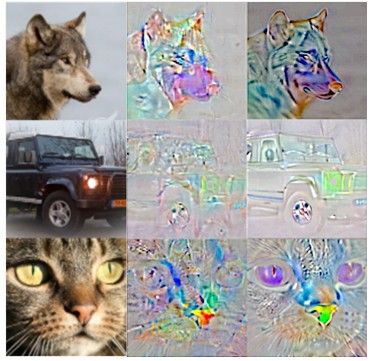 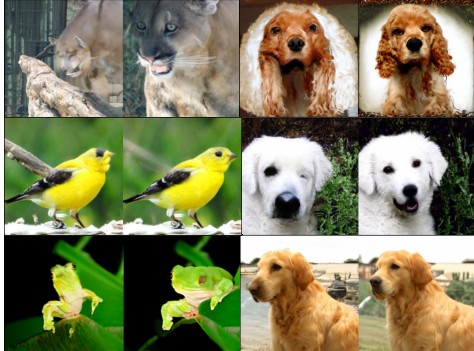

(a) PAG: Noisy-classifer vs DA-Classifier  (b) Generated Samples: Noisy-Classifier vs DA-Classifier

Figure 4: **(a)** Min-max normalized gradients on clean samples (left column) diffused to t = 300 (T = 999). For easy comparison between Noisy classifier gradients (middle column) and DA-classifier gradients (right column), we applied an identical enhancement to both images, i.e. contrast maximization. The unedited gradients are shown in Fig. 20. **(b)** Qualitative Comparison of Guidance Classifiers on the Image Generation Task using DDIM-100 with same random seed. In each pair, the first image is generated with the Noisy Classifier and the second image is generated with the Denoising-Augmented (DA) Classifier. We observe that the Denoising-Augmented (DA) Classifier improves overall coherence as compared to the Noisy Classifier. Also see Fig. 25 in appendix for more examples.

Table 5: Quantitative comparison of Guidance Classifiers on the Image Generation Task using 50k samples. We also show unconditional precision/recall (P/R) and the average class-conditional $\widetilde{\text{P}}$recision/$\widetilde{\text{R}}$ecall/$\widetilde{\text{D}}$ensity/$\widetilde{\text{C}}$overage.

| Method | CIFAR10 | | | | | | | | Imagenet | | | | |
|---|---|---|---|---|---|---|---|---|---|---|---|---|---|
| | FID↓ | IS↑ | P↑ | R↑ | $\widetilde{\text{P}}$↑ | $\widetilde{\text{R}}$↑ | $\widetilde{\text{D}}$↑ | $\widetilde{\text{C}}$↑ | FID↓ | sFID↓ | IS↑ | P↑ | R↑ |
| Noisy Classifier | 2.81 | 9.59 | 0.64 | 0.62 | 0.57 | 0.62 | 0.78 | 0.71 | 5.44 | **5.32** | 194.48 | 0.81 | 0.49 |
| DA-Classifier | **2.34** | **9.88** | 0.65 | 0.63 | **0.63** | **0.64** | **0.92** | **0.77** | **5.24** | 5.37 | **201.72** | 0.81 | 0.49 |

where they distinguish between noisy examples and their corresponding denoised examples as being in the ambient space and on the image manifold respectively, under certain assumptions. To determine the relative importance of noisy and denoised examples in DA-classifiers, we zeroed out one of the input images to the CIFAR10 classifier and measured classification accuracies: while zeroing the noisy input caused the average accuracy across all time-scales to drop to 50.1%, zeroing the denoised input breaks the classifier completely yielding random predictions.

**Classifier Gradients**. In Fig. 4a, we qualitatively compare between the noisy classifier gradients ($\nabla_{\mathbf{x}} \log p_\phi(y|\mathbf{x}, t)$) and the DA-classifier gradients($\nabla_{\mathbf{x}} \log p_\phi(y|\mathbf{x}, \hat{\mathbf{x}}, t)$). We find that the gradients obtained from the DA-classifier are more structured and semantically aligned with the clean image as compared to the ones obtained with the noisy classifier (see Figs. 20 and 22 in Appendix for more examples). Gradients backpropagated through the denoising score network have been previously utilized (e.g., [3, 10, 11, 18, 26, 36]), but our work is the first to observe and analyze the qualitative properties of gradients obtained by backpropagating through the denoising module (also see Fig. 21 in appendix).

**Image Generation.** To evaluate the guidance classifiers in terms of their image generation, we generate 50k images each – see Appendix C.2 for details on the sampling parameters. We compare the classifiers in terms of standard generative modeling metrics such as FID, IS, and P/R/D/C (Precision/Recall/Density/Coverage). The P/R/D/C metrics compare between the manifold of generated distribution and manifold of the source distribution in terms of nearest-neighbours and can be computed conditionally (i.e., classwise) or unconditionally. Following standard practice, we additionally evaluate CIFAR10 classifiers on class-conditional P/R/D/C. Our results (Table 5) show that our proposed Denoising-Augmented (DA) Classifier improves upon the Noisy Classifier in terms of FID and IS for both CIFAR10 and Imagenet at roughly same Precision and Recall levels (see Appendix C.3 for comparison with baselines). Our evaluation of average class-conditional precision, recall, density and coverage for each CIFAR10 class also shows that DA-classifiers outperform

Noisy classifiers: for example, DA-classifiers yield classwise density and coverage of about 0.92 and 0.77 respectively on average as compared to 0.78 and 0.71 obtained with Noisy-Classifiers. We can attribute our improvements in the class-conditional precision, recall, density and coverage to the improved generalization of DA-classifier. To qualitatively analyse benefits of the DA-classifier, we generated Imagenet samples using DDIM-100 sampler with identical random seeds and $\lambda_s = 2.5$. In our resulting analysis, we consistently observed that the DA-classifier maintains more coherent foreground and background as compared to the Noisy Classifier. We show examples in Fig. 4b. Overall, we attribute improved image generation to the improved generalization and classifier gradients.

# 6  Related Works

**Synthetic Augmentation with Diffusion Models** have also been explored to train semi-supervised classifiers [55] and few-shot learning [51]. Other studies on training classifiers with synthetic datasets generated with a text2image diffusion model include [4, 22, 56]. Apart from being computationally expensive, such text2image diffusion models are trained on large-scale upstream datasets and some of the reported improvements could also be attributed to the quality of the upstream dataset. Instead, we propose a efficient diffusion-based augmentation method and report improvements using a diffusion model trained with no extra data. Further, we also find that DiffAug is complementary to synthetic training data generated with large text2image diffusion models and leave further exploration of DiffAug with larger diffusion models as future work.

Synthetic examples have also been shown to be useful for enhancing training adversarially robust classifiers (e.g., [27, 53]) and extension of DiffAug for compute-efficient adversarial training is apt for exploration in future work.

**Diffusion Models for Robust Classification.** Diffusion-classifier [31] is a method for zero-shot classification but also improves robustness to covariate shifts. Diff-TTA[39] is a test-time adaptation technique to *update* the classifier parameters at test time and is complementary to classifier training techniques such as DiffAug. In terms of OOD detection, previous works have proposed reconstruction-based metrics for ood detection [20, 32]. To the best of our knowledge, this work is the first to demonstrate improved OOD detection on ImageNet-1k using diffusion models.

# 7  Conclusion

In this work, we introduce DiffAug to train robust classifiers with one-step diffusion denoised examples. The simplicity and computational efficiency of DiffAug enables us to also extend other data augmentation techniques, where we find that DiffAug confers additional robustness without affecting accuracy on clean examples. We qualitatively analyse DiffAug samples in an attempt to explain improved robustness. Furthermore, we extend DiffAug to test time and introduce an efficient test-time image adaptation technique to further improve robustness to covariate shifts. Finally, we theoretically analyse perceptually aligned gradients in denoised examples and use this to improve classifier-guided diffusion. Overall, we present effective augmentation technique using diffusion models trained with no external datasets.

**Acknowledgements**
We thank the Canadian Institute for Advanced Research (CIFAR) for their support. Resources used in preparing this research were provided, in part, by NSERC, the Province of Ontario, the Government of Canada through CIFAR, and companies sponsoring the Vector Institute `www.vectorinstitute.ai/#partners`. We thank Kry Yik Lui for their feedback and suggestion to explore connections with denoised-smoothing. We thank Scott Lowe, Emily Napier, Marvin F. Silva and Felix Dangel for providing feedback on our figures/terminologies. Chandramouli Sastry is thankful to the teachings of Shankaracharyas and Guru Parampara for instilling the values of enquiry, patience and perseverence. Chandramouli Sastry is also supported by Borealis AI Fellowship. We sincerely thank the anonymous reviewers and area-chairs for their service and valuable feedback & suggestions to strengthen our paper.

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

# Part I

# Appendix

## Table of Contents

**Contribution Statement**

CSS and SHD are student contributors of this project and their contributions are as follows: CSS proposed the key research idea, designed and carried out all the experiments in this paper, worked on the theoretical derivations and wrote the first draft of this paper. SHD provided advice and constructive criticisms throughout the research and development phase.

**Limitations and Future Work**

In this work, we introduced DiffAug, a simple diffusion-based augmentation technique to improve classifier robustness. While we presented a unified training scheme, we follow previous works and evaluate robustness to covariate shifts, adversarial examples and out-of-distribution detection using separate evaluation pipelines (for e.g., we do not apply DDA on OOD examples). A unified evaluation pipeline for classifier robustness is an open problem and out of scope for this paper. The key advantage of our method is its computational efficiency since it requires just one reverse diffusion step to generate the augmentations; however, this is computationally more expensive than handcrafted augmentation techniques such as AugMix. Nevertheless, it is fast enough that we can generate the augmentations online during each training step. With recent advances in distilling diffusion models for fast sampling [34] such as Consistency-Models [48], it may be possible to generate better quality synthetic examples within the training loop. Since the improved robustness introduced by DiffAug can be attributed to the augmentations of varying image quality (Fig. 1), we believe that this complementary regularization effect can still be valuable with efficient high-quality sampling and leave further exploration to future work. Likewise, DiffAug is complementary to the use of additional synthetic data generated offline with text2image diffusion models and the extension of DiffAug to text2image diffusion models is suitable for future investigation. We also extend DiffAug to test time and propose DE, wherein we demonstrate improvements comparable to DDA at 10x wallclock time on all classifiers except ViT on Imagenet-C; while this demonstrates a limitation of single-step denoising (i.e., DE) vs. multi-step denoising(i.e., DDA), we also note that DE improves over DDA for all classifiers when considering Imagenet-R and Imagenet-S. While we demonstrate improvements over existing baselines of classifier-guidance (e.g., [6, 29, 58]), we do not compare with classifier-free guidance [25], the popular guidance method, since our primary focus is on the training of robust classifiers with DiffAug and demonstrating the potential of DiffAug. Training classifier-guided diffusion models for careful comparison requires additional computational resources and we leave this analysis to future work. Nevertheless, classifier-guidance is a computationally

attractive alternative to perform conditional generation since this allows flexible reuse of a pretrained diffusion model for different class-definitions by separately training a small classifier model. We also note that computing classifier-gradients is slower for DA-classifiers as compared to noisy classifiers since it requires backpropagating through the score-network and shares this limitation with other recent works that utilize intermediate denoised examples for guidance (e.g., Bansal et al. [3], Chung et al. [10, 11], Ho et al. [26]) – the recent advances in efficient diffusion sampling could be extended to class-conditional sampling to reduce the gap.

**Compute Resources**

For this paper, we had access to 8 40GB A40 GPUs to conduct our training and evaluation. We used a maximum of 4 GPUs for each job and the longest training job was the 90-epoch RN-50 training followed by the 20-epoch ViT training. The evaluation of classifiers on Imagenet-C, Imagenet-R and Imagenet-S using DE and Default evaluation modes are fairly fast. However, generating DDA examples for the entire Imagenet-C dataset and Imagenet-test dataset is computationally expensive and takes up to a week (even while using 8 GPUs in parallel) and also requires sufficient storage capacity to save the DDA-transformed images. Likewise, evaluation of certified accuracy with DDS is also computationally expensive since it uses 10k noise samples per example to estimate the certified radius and prediction. Training CIFAR10 guidance classifiers are fairly efficient while finetuning the Imagenet guidance classifier can take up to 3 days depending on the gradient accumulation parameter (i.e., number of GPUs available) – when available, we used a maximum of 4 GPUs. On average, the 50k CIFAR10 and Imagenet images that we sample for evaluation of guidance classifiers can be done within a maximum of 36 hours depending on how we parallelize the generation.

**Broader Impact**

The main contribution of this paper is to introduce a new augmentation method using diffusion models to improve classifier robustness. With increasing deployment of deep learning models in real-world settings, improved robustness is crucial to enable safe and trustworthy deployment. Since we demonstrate potential of diffusion models trained with no extra data as compared to the classifier, we anticipate that this will be useful in applications where such extra data is not easily available (e.g., medical imaging). We also extend this method to improve classifier-guided diffusion. Improvements in classifier-guided diffusion can be used in developing a myriad downstream applications, each with their own potential balance of positive and negative impacts. Leveraging and re-using a pre-trained model amplifies the importance of giving proper consideration to copyright issues associated with the data on which that model was trained. While our proposed model has the potential for generating deep-fakes or disinformation, these technologies also hold promise for positive applications, including creativity-support tools and design aids in engineering.

# A    Appendix for Section 3

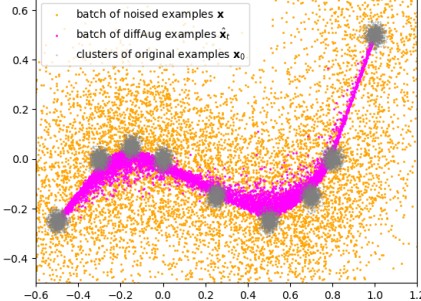

Figure 5: A demonstration of the DiffAug technique using a Toy 2D dataset.

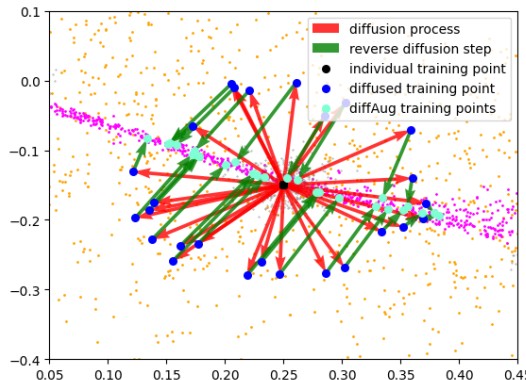

Figure 6: A zoomed-in view demonstrating the transformation considering a single train point.

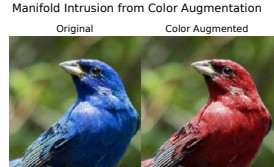

Figure 7: Example of Manifold Intrusion from Appendix C of Hendrycks et al. [24]. While DiffAug may alter class labels (Fig. 1), the denoised images are visually distinguishable from the original images allowing the model to also *learn* from noisy labels without inducing manifold intrusion. On the other hand, here is an example of manifold intrusion where the augmented image does not contain any visual cues that enable the model to be robust to noisy labels.

## B Appendix for Section 4

### B.1 Training Details

In the following, we describe the training details for the classifiers we evaluated in Section 4. In general, we optimize a sum of two losses: $\mathcal{L}_{\text{Total}} = \mathcal{L}_{\text{Orig}} + \mathcal{L}$ where, $\mathcal{L}_{\text{Orig}}$ is the classification objective on the original augmentation policy that we aim to improve using DiffAug examples and $\mathcal{L}$ denotes the classification objective measured on DiffAug examples (Eq. (5)). Before applying DiffAug, we first resized the raw image such that at least one of the edges is of size 256 and then use a $224 \times 224$ center-crop as the test image since the diffusion model was not trained on random resized crops.

- RN-50: We trained the model from scratch for 90 epochs using the same optimization hyperparameters used to train the official PyTorch RN-50 checkpoint.

- ViT: We used the two-stage training recipe proposed in DeIT-III. In particular, the training recipe consists of an 800-epoch supervised pretraining at a lower resolution (e.g., $192 \times 192$) followed by a 20-epoch finetuning at the target resolution (e.g., $224 \times 224$). Starting with the pretrained checkpoint (i.e., after 800 epochs), we finetune the classifier exactly following the prescribed optimization and augmentation hyperparameters (e.g., AutoAugment (AA) parameters and MixUp/CutMix parameters) except that we also consider DiffAug examples. We also included DiffAug examples when applying MixUp/CutMix since we observed significant drops in standard test accuracies when training directly on DiffAug examples without label-smoothing or MixUp. We briefly explored stacking DiffAug with AA and identified that this did not introduce any noticeable change as compared to independent application of DiffAug and AA to train examples.

- AugMix/DeepAugment/DeepAugment+AugMix: To evaluate the combination of DiffAug with these augmentations, we finetune the RN-50 checkpoint opensourced by the respective papers for 10 epochs with a batch-size of 256. We resume the training with the optimizer state made available along with the model weights and use a constant learning rate of 1e-6.

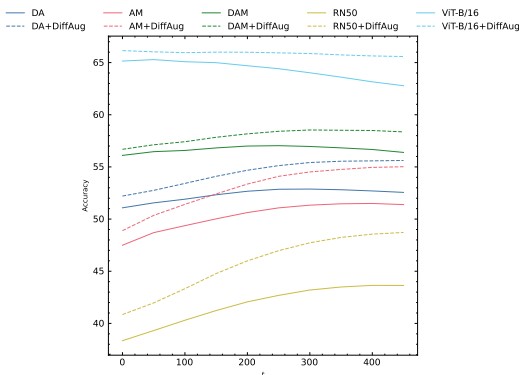

Figure 8: ImageNet-C Accuracy (Top-1) averaged across all severities. While $t = 0$ corresponds to the accuracy in default evaluation mode, other values of $t$ correspond to the DiffAug-Ensemble accuracy. See Fig. 11(a) for the corresponding graph on ImageNet-C (severity=5).

Table 6: ImageNet-C (severity=5) accuracy for each corruption type. Relative Improvements when additionally using diffusion denoised augmentations are computed with respect to the corresponding pretrained checkpoints and averaged across all corruption types. Overall, we observe improvements for each family of corruptions: in the default evaluation mode, we observe an average absolute improvement of 2.5%, 4.9%,1.0% and 0.76% for the Noise, Blur, Weather and Digital corruptions respectively. Across all evaluation modes, we observe an average absolute improvement of 1.86%, 4.74%, 1.67%, 1.51% for the Noise, Blur, Weather and Digital corruptions respectively.

| Inference Mode. | Train Aug. | Noise | | | Blur | | | | Weather | | | | Digital | | | | Avg. | Rel. Imp. |
|---|---|---|---|---|---|---|---|---|---|---|---|---|---|---|---|---|---|---|
| | | Gauss. | Shot | Impul. | Defoc. | Glass | Motion | Zoom | Snow | Frost | Fog | Brit. | Contr. | Elastic | Pixel | JPEG | | |
| DDA | AM | 50.64 | 52.22 | 50.8 | 18.18 | 25.2 | 18.76 | 24.29 | 22.67 | 33.04 | 5.35 | 40.56 | 11.12 | 40.68 | 54.15 | 50.08 | 33.18 | 0 |
| | AM+DiffAug | 51.3 | 52.64 | 51.81 | 21.14 | 27.75 | 23.02 | 28.09 | 23.13 | 34.61 | 7.42 | 40.79 | 11.35 | 41.03 | 55.15 | 50.34 | 34.64 | 8 |
| | DA | 51.48 | 53.37 | 51.15 | 24.11 | 30.09 | 18.06 | 23.24 | 25.31 | 35.46 | 10.69 | 44.13 | 12.33 | 41.28 | 58.66 | 51.84 | 35.41 | 0 |
| | DA+DiffAug | 52.21 | 53.96 | 51.74 | 29.26 | 33.09 | 23.06 | 27.84 | 25.47 | 36.98 | 13.66 | 47.09 | 14.51 | 41.84 | 60.63 | 52.85 | 37.61 | 9.75 |
| | DAM | 53.5 | 55.56 | 54.86 | 31.35 | 37.44 | 28.91 | 28.52 | 30.76 | 39.67 | 12.88 | 49.06 | 21.28 | 45.04 | 61.63 | 54.95 | 40.36 | 0 |
| | DAM+DiffAug | 54.15 | 55.96 | 55.33 | 33.47 | 38.2 | 33.5 | 31.99 | 31.24 | 41.09 | 15.53 | 50.91 | 23.42 | 44.77 | 62.99 | 56.14 | 41.91 | 5.52 |
| | RN50 | 45.95 | 47.62 | 46.72 | 12.89 | 17.98 | 12.77 | 20.29 | 17.97 | 27.56 | 5.11 | 35.42 | 5.91 | 36.1 | 47.91 | 45.06 | 28.35 | 0 |
| | RN50+DiffAug | 47.46 | 48.56 | 48.16 | 17.65 | 22.7 | 18.16 | 23.47 | 19.05 | 30.35 | 7.89 | 38.11 | 6.86 | 37.51 | 52.9 | 48.37 | 31.15 | 16.35 |
| | ViT-B/16 | 54.6 | 55.75 | 54.79 | 32.65 | 40.41 | 33.19 | 30.38 | 36.25 | 42.3 | 21.91 | 51.9 | 28.7 | 48.53 | 64.01 | 58.7 | 43.6 | 0 |
| | ViT-B/16+DiffAug | 56.37 | 57.18 | 56.45 | 32.79 | 42.13 | 35.76 | 35.03 | 36.65 | 43.25 | 21.49 | 55.15 | 26.54 | 49.78 | 66.2 | 61.05 | 45.05 | 3.12 |
| DDA-SE | AM | 49.59 | 51.23 | 50.07 | 20.61 | 23.03 | 24.43 | 32.79 | 25.76 | 36.26 | 20.05 | 54.14 | 14.16 | 39.15 | 54.62 | 52.24 | 36.54 | 0 |
| | AM+DiffAug | 51.41 | 52.34 | 51.79 | 24.45 | 27.2 | 29.82 | 36.97 | 26.32 | 37.4 | 26.01 | 53.67 | 15.49 | 39.08 | 54.96 | 52.17 | 38.61 | 8.32 |
| | DA | 53.36 | 54.71 | 53.34 | 25.72 | 28.22 | 20.1 | 26.37 | 30.57 | 40.11 | 28.77 | 59.39 | 13.91 | 39.82 | 59.1 | 52.36 | 39.06 | 0 |
| | DA+DiffAug | 53.92 | 55.37 | 53.98 | 30.9 | 30.45 | 25.19 | 30.78 | 31.01 | 41.19 | 35.38 | 60.76 | 18.72 | 39.75 | 59.98 | 52.24 | 41.31 | 9.24 |
| | DAM | 54.17 | 56.31 | 55.19 | 33.61 | 34.12 | 36.34 | 34.87 | 35.82 | 45.12 | 35.52 | 60.9 | 27.85 | 43.33 | 62.99 | 56 | 44.81 | 0 |
| | DAM+DiffAug | 54.53 | 56.83 | 55.96 | 36.19 | 35.73 | 40.5 | 37.72 | 36.39 | 45.96 | 39.19 | 61.89 | 31.84 | 42.59 | 63.43 | 56.56 | 46.35 | 4.32 |
| | RN50 | 44.85 | 45.59 | 45.17 | 14.33 | 16.2 | 14.23 | 23.96 | 20.54 | 30.4 | 19.56 | 51.67 | 6.61 | 33.2 | 46.45 | 46.56 | 30.62 | 0 |
| | RN50+DiffAug | 47.34 | 48.48 | 48.11 | 21.26 | 21.49 | 21.84 | 28.09 | 20.98 | 31.96 | 20.98 | 51.72 | 7.26 | 34.02 | 51.05 | 48.06 | 33.51 | 14.01 |
| | ViT-B/16 | 58.41 | 58.73 | 58.58 | 36.58 | 38.17 | 41.8 | 36.45 | 52.79 | 58.7 | 58.95 | 70.05 | 47.41 | 48.19 | 65.2 | 63.45 | 52.9 | 0 |
| | ViT-B/16+DiffAug | 59.11 | 59.55 | 59.07 | 37.2 | 40.35 | 43.26 | 41.33 | 52.68 | 57 | 55.53 | 70.36 | 48.03 | 48.23 | 66.54 | 64.9 | 53.54 | 1.66 |
| DE | AM | 32.52 | 35.16 | 33.01 | 21.02 | 31.16 | 25.57 | 32.64 | 25.83 | 38.74 | 16.89 | 54.76 | 7.17 | 42.98 | 54.85 | 58.89 | 34.08 | 0 |
| | AM+DiffAug | 37.62 | 39.08 | 36.38 | 28.96 | 37.44 | 35.66 | 40.95 | 27.34 | 40.48 | 26.67 | 56.3 | 9.4 | 43.93 | 58.21 | 60.25 | 38.58 | 18.17 |
| | DA | 44.96 | 45.75 | 46.13 | 20.14 | 30.21 | 22.41 | 29.16 | 29.81 | 39.7 | 23.93 | 57.59 | 4.5 | 43.9 | 57.15 | 60.94 | 37.08 | 0 |
| | DA+DiffAug | 45.65 | 46.28 | 46.65 | 27.91 | 35.08 | 29.37 | 35.71 | 30.79 | 41.9 | 32.69 | 59.93 | 10.17 | 43.71 | 58.86 | 61.64 | 40.42 | 19.42 |
| | DAM | 46.43 | 48.4 | 47.24 | 28.15 | 37.9 | 32.41 | 35.55 | 34.83 | 44.27 | 28.48 | 59.69 | 15.43 | 46.14 | 60.77 | 62.23 | 41.86 | 0 |
| | DAM+DiffAug | 48.06 | 49.68 | 48.52 | 33.04 | 41.36 | 38.84 | 40.22 | 35.97 | 45.71 | 36.16 | 61.01 | 22.15 | 46.27 | 61.83 | 62.79 | 44.77 | 10.04 |
| | RN50 | 26.63 | 28 | 27.23 | 11.76 | 18.91 | 16.26 | 24.85 | 20.3 | 31.65 | 15.29 | 49.09 | 1.61 | 36.51 | 45.63 | 53.13 | 27.12 | 0 |
| | RN50+DiffAug | 31.73 | 33.42 | 31.93 | 18.38 | 27.77 | 23.82 | 32.91 | 21.77 | 34.37 | 25.01 | 52.52 | 2.48 | 38.61 | 51.41 | 57.2 | 32.22 | 26.99 |
| | ViT-B/16 | 54.7 | 52.21 | 55.16 | 29.2 | 38.02 | 36.58 | 36.26 | 51.95 | 58.6 | 41.89 | 70.41 | 18.9 | 50.35 | 61.64 | 67.87 | 48.25 | 0 |
| | ViT-B/16+DiffAug | 57.22 | 54.87 | 57.88 | 33.95 | 41.05 | 43.11 | 45.46 | 49.74 | 58.58 | 44.93 | 72.75 | 30.38 | 51.27 | 67.04 | 69.81 | 51.87 | 10.84 |
| Def. | AM | 15.01 | 18.38 | 16.64 | 21.48 | 13.69 | 24.89 | 33.67 | 21.54 | 27.13 | 22.91 | 57.92 | 13.08 | 25.17 | 42.32 | 46.99 | 26.72 | 0 |
| | AM+DiffAug | 19.5 | 22.33 | 20.58 | 26.34 | 17.88 | 31.11 | 37.9 | 22.53 | 28.21 | 28.76 | 56.98 | 15.24 | 24.62 | 43.02 | 47.09 | 29.47 | 14.3 |
| | DA | 39.61 | 40.8 | 41.89 | 25.48 | 15.74 | 19.01 | 24.58 | 27.42 | 33.58 | 32.04 | 62.61 | 9.55 | 23.69 | 45.41 | 37.48 | 31.93 | 0 |
| | DA+DiffAug | 40.79 | 41.76 | 43.15 | 31.52 | 17.58 | 23.6 | 28.54 | 27.67 | 34.93 | 37.27 | 63.11 | 15 | 23.11 | 44.38 | 34.33 | 33.78 | 10 |
| | DAM | 39.61 | 42.75 | 42.14 | 34.47 | 22.95 | 36.57 | 35.58 | 34.04 | 39.85 | 38.75 | 63.95 | 25.6 | 29.62 | 56.44 | 50.51 | 39.52 | 0 |
| | DAM+DiffAug | 40.86 | 44.04 | 43 | 37.08 | 26.04 | 40.67 | 37.77 | 35.17 | 40.67 | 41.24 | 64.25 | 31.13 | 28.8 | 57.06 | 50.88 | 41.24 | 5.3 |
| | RN50 | 5.69 | 6.49 | 6.45 | 15.04 | 8.23 | 13.29 | 22.85 | 15.59 | 20.44 | 22.22 | 55.64 | 4.23 | 14.31 | 23 | 34.55 | 17.87 | 0 |
| | RN50+DiffAug | 9.51 | 10.4 | 10.71 | 23.08 | 14.01 | 21.06 | 28.4 | 15.75 | 21 | 22.95 | 54.56 | 4.16 | 15.95 | 25.76 | 35.8 | 20.87 | 28.68 |
| | ViT-B/16 | 51.78 | 48.67 | 51.92 | 37.13 | 19.69 | 43.04 | 36.99 | 55.93 | 60.76 | 69.17 | 74.79 | 56.76 | 35.22 | 56.95 | 62.38 | 50.75 | 0 |
| | ViT-B/16+DiffAug | 54.12 | 50.53 | 54.04 | 41.11 | 30.52 | 45.43 | 42.75 | 55.06 | 60.88 | 70.21 | 74.92 | 56.96 | 33.88 | 58.14 | 63.22 | 52.78 | 6.64 |

Table 7: Top-1 Accuracy (%) on Imagenet-S and Imagenet-R. We summarize the results for each combination of Train-augmentations and evaluation modes. The average (avg) accuracies for each classifier and evaluation mode is shown.

| Train Augmentations | ImageNet-S | | | | | ImageNet-R | | | | |
|---|---|---|---|---|---|---|---|---|---|---|
| | DDA | DDA (SE) | DE | Def. | **Avg** | DDA | DDA (SE) | DE | Def. | **Avg** |
| AM | 11.53 | 13.74 | 15.73 | 11.17 | 13.04 | 36.39 | 42.02 | 42.22 | 41.03 | 40.42 |
| AM+DiffAug | 12.30 | 14.02 | 16.00 | 10.95 | **13.32** | 37.09 | 42.14 | 43.28 | 40.98 | **40.87** |
| DA | 12.18 | 15.69 | 17.16 | 13.84 | 14.72 | 37.82 | 43.11 | 43.50 | 42.24 | 41.67 |
| DA+DiffAug | 13.04 | 16.28 | 17.80 | 14.06 | **15.30** | 38.83 | 43.62 | 44.39 | 42.61 | **42.36** |
| DAM | 14.63 | 19.68 | 20.05 | 19.47 | 18.46 | 41.47 | 46.64 | 46.25 | 46.78 | 45.29 |
| DAM+DiffAug | 15.41 | 20.00 | 20.22 | 19.82 | **18.86** | 42.37 | 47.12 | 46.67 | 47.05 | **45.80** |
| RN50 | 9.38 | 10.29 | 11.68 | 7.12 | 9.62 | 32.85 | 38.24 | 38.49 | 36.16 | 36.44 |
| RN50+DiffAug | 10.25 | 10.63 | 12.52 | 6.99 | **10.10** | 34.76 | 39.65 | 41.61 | 37.55 | **38.39** |
| ViT-B/16 | 16.22 | 24.40 | 23.97 | 25.36 | 22.49 | 44.62 | 53.84 | 53.30 | 53.61 | 51.34 |
| ViT-B/16+DiffAug | 18.86 | 25.14 | 24.77 | 25.74 | **23.63** | 47.71 | 55.36 | 55.80 | 54.98 | **53.46** |
| **Avg** | 13.38 | 16.99 | 17.99 | 15.45 | 15.95 | 39.39 | 45.17 | 45.55 | 44.30 | 43.60 |
| **Avg (No-DiffAug)** | 12.79 | 16.76 | 17.72 | 15.39 | 15.66 | 38.63 | 44.77 | 44.75 | 43.96 | 43.03 |
| **Avg (DiffAug)** | 13.97 | 17.21 | 18.26 | 15.51 | 16.24 | 40.15 | 45.58 | 46.35 | 44.63 | 44.18 |

Table 8: Top-1 Accuracy (%) across different types of distribution shifts when additional high-quality synthetic data from Stable-Diffusion is available (denoted by +Synth). For reference, we also include the results for the corresponding ResNet50 models without extra synthetic data.

| Model | ImageNet-Test | | ImageNet-C (Severity=5) | | ImageNet-R | | ImageNet-S | | ImageNet Sketch | | ImageNet-A | | ImageNet-D | | Average | | |
|---|---|---|---|---|---|---|---|---|---|---|---|---|---|---|---|---|---|
| | DE | Def. | DE | Def. | DE | Def. | DE | Def. | DE | Def. | DE | Def. | DE | Def. | DE | Def | Total |
| RN50 | 71.43 | 76.15 | 27.12 | 17.87 | 38.49 | 36.16 | 11.68 | 7.12 | 25.27 | 24.09 | 0.77 | 0.03 | 13.14 | 11.36 | 26.84 | 24.68 | 25.76 |
| RN50+DiffAug | 75.07 | 75.95 | 32.22 | 20.87 | 41.61 | 37.55 | 12.52 | 6.99 | 26.67 | 24.8 | 1.09 | 0.56 | 11.37 | 10.37 | **28.65** | **25.30** | **26.97** |
| RN50+Synth | 68.83 | 75.47 | 25.05 | 17.58 | 50.37 | 49.28 | 11.83 | 7.68 | 36.41 | 35.45 | 1.32 | 0.63 | 21.06 | 17.52 | 30.70 | 29.09 | 29.89 |
| RN50+Synth+DiffAug | 73.85 | 75.01 | 30.06 | 20.2 | 54.71 | 50.85 | 13.57 | 7.92 | 37.39 | 35.33 | 1.53 | 1.12 | 21.41 | 19.18 | **33.22** | **29.94** | **31.58** |

Table 9: Top-1 Accuracy (%) on Imagenet-A and Imagenet-D. We summarize the results for each combination of train-augmentations and evaluation modes. The average (avg) accuracies for each classifier and evaluation mode is shown.

| Train Augmentations | ImageNet-A | | | ImageNet-D | | |
|---|---|---|---|---|---|---|
| | DE | Def | **Avg** | DE | Def | **Avg** |
| AM | 3.00 | 3.67 | 3.34 | 12.43 | 11.16 | **11.80** |
| AM+DiffAug | 3.47 | 4.09 | **3.78** | 11.94 | 11.33 | 11.64 |
| DA | 3.09 | 3.39 | 3.24 | 13.19 | 11.2 | **12.20** |
| DA+DiffAug | 3.45 | 3.52 | **3.49** | 13.03 | 11.07 | 12.05 |
| DAM | 3.15 | 3.84 | 3.50 | 14.64 | 12.11 | **13.38** |
| DAM+DiffAug | 3.44 | 4.43 | **3.94** | 12.78 | 11.73 | 12.26 |
| RN50+Synth | 1.32 | 0.63 | 0.98 | 21.06 | 17.52 | 19.29 |
| RN50+Synth+DiffAug | 1.53 | 1.12 | **1.33** | 21.41 | 19.18 | **20.30** |
| RN50 | 0.77 | 0.03 | 0.40 | 13.14 | 11.36 | **12.25** |
| RN50+DiffAug | 1.09 | 0.56 | **0.83** | 11.37 | 10.77 | 11.07 |
| VIT | 25.88 | 39.81 | 32.85 | 17.65 | 16.4 | 17.03 |
| VIT+DiffAug | 27.6 | 39.35 | **33.48** | 17.94 | 16.61 | **17.28** |
| **Avg** | 6.48 | 8.70 | 7.59 | 15.05 | 13.37 | 14.21 |
| **Avg (No-DiffAug)** | 6.20 | 8.56 | 7.38 | 15.35 | 13.29 | 14.32 |
| **Avg (DiffAug)** | 6.76 | 8.85 | 7.80 | 14.75 | 13.45 | 14.10 |

Table 10: DDA vs DE in terms of wallclock times: We use 40GB A40 GPU for determining the running time. For each method, we determine the maximum usable batch-size and report the average wallclock time for processing a single example.

| Method | Wallclock Time (s) |
|---|---|
| DE | 0.5 |
| DDA | 4.75 |

### B.2.1 Extra Synthetic Data with ImageNet-only Diffusion models

In this section, we conduct a preliminary analysis of extending training data with fully-synthesized examples generated with a diffusion model trained only on ImageNet data. In particular, we generate 50k new examples using 4 different techniques as shown in Table 11. From these experiments, we find that fully synthesized examples from ImageNet-only diffusion models do not enhance the robustness as compared to the synthetic examples from stable-diffusion. Overall, we find that DiffAug uses an ImageNet-only diffusion-model to achieve robustness improvements although the fully-synthesized examples from the same diffusion models do not enhance robustness.

Table 11: An evaluation of extra synthetic data with ImageNet-only diffusion models. We generate 50k new examples to extend the ImageNet train-dataset and finetune the torchvision resnet-50 models for 10 epochs.

| Diffusion Model | Guidance Method | Sampler | Steps | FID | IS | ImageNet Test | ImageNet-C | | ImageNet-R | |
|---|---|---|---|---|---|---|---|---|---|---|
| | | | | | | | Def | DE | Def | DE |
| Improved DDPM (Nichol and Dhariwal [35]) | Classifier-Guidance [Noisy Classifier] | DDPM | 250 | 3.94 | 215.84 | 75.53 | 17.91 | 27.41 | 36.13 | 38.32 |
| | Classifier-Guidance [Noisy Classifier] | DDIM | 25 | 5.44 | 194.48 | 75.68 | 17.84 | 27.34 | 36.02 | 38.38 |
| | Classifier-Guidance [DA Classifier] | DDIM | 25 | 5.24 | 201.72 | 75.64 | 17.97 | 27.31 | 36.27 | 38.59 |
| DiT-XL (Peebles and Xie [37]) | Classifier-Free Guidance | DDPM | 250 | 2.24 | 279.91 | 75.68 | 17.91 | 27.19 | 36.68 | 38.63 |
| RN50 | | | | | | 76.15 | 17.87 | 27.12 | 36.16 | 38.49 |
| RN50+DiffAug | | | | | | 75.95 | 20.87 | 32.22 | 37.55 | 41.61 |

### B.2.2 DiffAug with Other Samplers

The single reverse-diffusion step used in DiffAug can be understood as a single reverse-diffusion step of the DDIM [46] sampler. Alternatively, we can consider improved diffusion samplers and in this section, we explore DPM-Solver[33]. We illustrate the augmentations in Fig. 9.

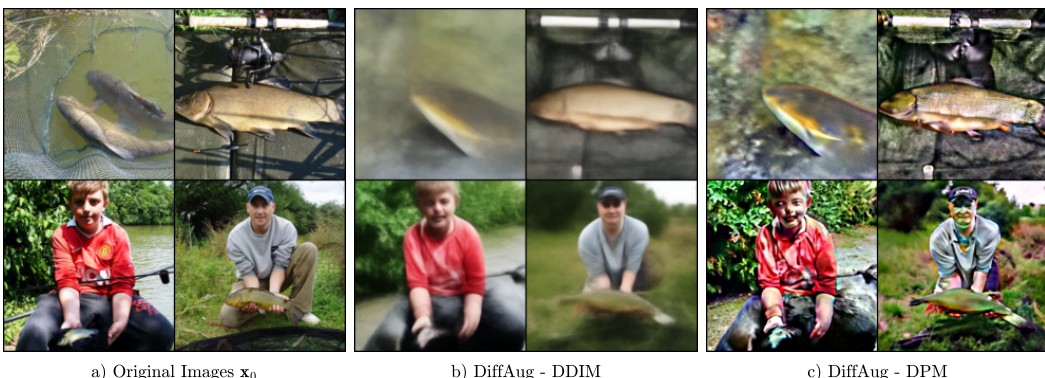

a) Original Images $\mathbf{x}_0$  b) DiffAug - DDIM  c) DiffAug - DPM

Figure 9: An illustration of DiffAug with DDIM and DPM solvers: we show DiffAug augmentations at $t = 500$ for the examples in (a) applied using one reverse-diffusion step of the DDIM sampler (b) and the DPM-solver (c).

We evaluate a single-step of reverse-diffusion of DPM-solver-2, both at train-time as well as at test-time. In particular, we first diffused the train example to a random diffusion-time $t$ and used a single reverse-diffusion step of the order-2 DPM solver to integrate the diffusion ODE backwards from diffusion time $t$ to $t = \epsilon$ with $\epsilon = 0.01$ instead of $\epsilon = 0$ for numerical stability. Based on the results shown in Table 12, DDIM appears to be the better sampling strategy for DiffAug although the resulting augmented images are somewhat of high visual quality when using DPM-Solver. While DiffAug using DPM-Solver may not be beneficial for classification, there may be other applications of such augmentations and leave further exploration to future work.

Table 12: Evaluation of other sampling methods for DiffAug: We use ↓ to denote lower performance due to the use of DPM-Solver instead of DDIM.

| Training Method | Evaluation on ImageNet-C (severity=5) | | |
|---|---|---|---|
| | Default | DiffAug-Ensemble (DDIM) | DiffAug-Ensemble (DPM-Solver-2) |
| AM | 26.72 | 34.08 | 31.77↓ |
| AM+DiffAug/DDIM | 29.47 | 38.58 | 35.56↓ |
| AM+DiffAug/DPM-Solver-2 | 22.96↓ | 29.69↓ | 25.16 |

## B.3 Certified Accuracy Experiments

We follow DDS[5] and previous works on denoised smoothing and evaluate the certified accuracy on a randomly selected subset of 1k Imagenet samples. The classifiers are generally evaluated with 3 noise scales: $\sigma_t \in \{0.25, 0.5, 1.0\}$ and for each $l_2$ radius and model pair, the noise that yields the best certified accuracy at that radius is selected and summarized in Table 13, following previous works. We also show the certified accuracy plots for each Gaussian perturbation separately in Fig. 10.

| | Certified Accuracy (%) at $l_2$ radius. | | | | | |
|---|---|---|---|---|---|---|
| | 0.5 | 1.0 | 1.5 | 2.0 | 2.5 | 3.0 |
| ViT | 36.30 | 25.50 | 16.72 | 14.10 | 10.70 | 8.10 |
| ViT+DiffAug | **40.30** | **32.50** | **23.62** | **19.40** | **15.20** | **11.00** |

Table 13: Certified Accuracy for different $l_2$ perturbation radius. As is standard in the literature, we consider $\sigma_t \in \{0.25, 0.5, 1.0\}$ and select the best $\sigma_t$ for each $l_2$ radius.

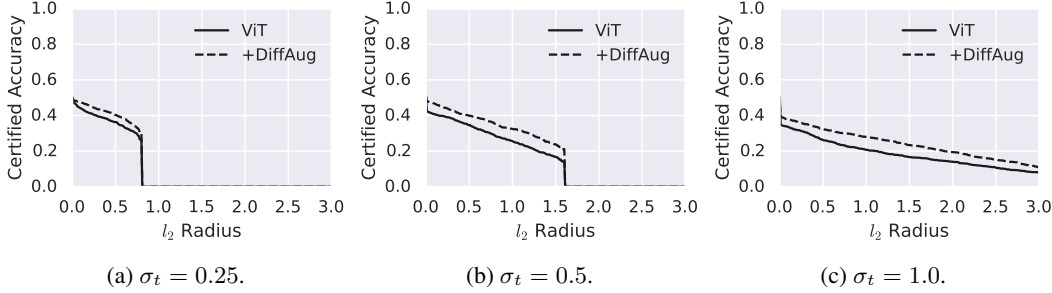

(a) $\sigma_t = 0.25$.  (b) $\sigma_t = 0.5$.  (c) $\sigma_t = 1.0$.

Figure 10: $l_2$ Radius vs Certified Accuracy for different values of $\sigma_t$.

### B.3.1 DiffAttack Adversarial Examples

We also explored robustness to DiffAttack[9] adversarial examples: these are unrestricted adversarial examples generated with a stable-diffusion model. Here, we generate adversarial examples using their official code: in particular, we use the inception-v3 to generate the adversarial examples and evaluate the other classifiers. We tabulate the results in Table 14.

Table 14: DiffAttack Evaluation: We generate DiffAttack adversarial examples for the ImageNet-compatible dataset and evaluate the robustness enhancements introduced by DiffAug training and DiffAug-Ensemble inference. We consistently observe that DiffAttack succeeds in attacking all models albeit to different extents. In each of these cases, DiffAug-training followed by DE inference enhances the performance. Most notably, ViT+DiffAug/DE comes closest to the original performance.

| Training Method | ImageNet-Compatible | | | |
| --- | --- | --- | --- | --- |
| | Original | Adversarial | | |
| | Def | Def | DE | DiffAug/DE Improvement |
| DA | 91.80 | 66.20 | 68.10 | |
| DA+DiffAug | 91.70 | 66.40 | 72.40 | 6.20 |
| AM | 92.80 | 66.80 | 69.50 | |
| AM+DiffAug | 92.50 | 67.20 | 73.20 | 6.40 |
| DAM | 92.70 | 71.50 | 72.80 | |
| DAM+DiffAug | 91.70 | 70.60 | 74.60 | 3.10 |
| ViT | 95.70 | 78.70 | 77.80 | |
| ViT+DiffAug | **96.40** | 80.80 | **87.20** | 8.50 |
| RN50 | 92.70 | 61.10 | 62.10 | |
| RN50+DiffAug | 91.90 | 60.20 | 71.00 | 9.90 |
| RN50+Synth | 92.20 | 58.60 | 58.90 | |
| RN50+Synth+DiffAug | 90.80 | 59.80 | 68.70 | 10.10 |

## B.4 Detailed OOD Detection Results

OOD Detection results are mainly evaluated with with two metrics: AUROC and FPR@TPR95. The AUROC is a threshold-free evaluation of OOD detection while FPR@TPR95 measures the false positive rate at which OOD samples are incorrectly identified as in-distribution samples given that the true positive rate of detecting in-distribution samples correctly is 95%. The Near-OOD Imagenet task as defined by OpenOOD consists of SSB-Hard and NINCO datasets and we also include the performance for each dataset in the following tables.

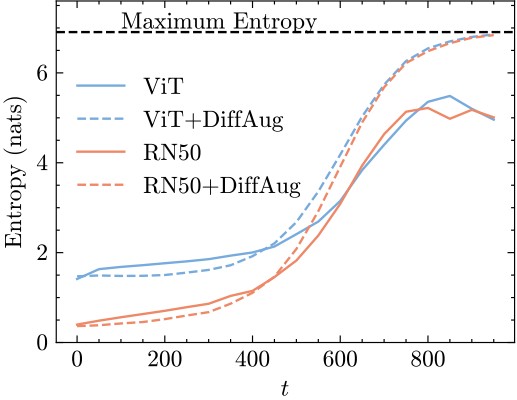

Figure 11: The entropy plots for ViT and RN50 are shown where we observe similar trends as in Fig. 2.

Table 15: AUROC and FPR (lower is better) on ImageNet Near-OOD Detection.

| Train Augmentation | AUROC | | | | | FPR95 | | | | |
|---|---|---|---|---|---|---|---|---|---|---|
| | ASH | MSP | ReAct | SCALE | Avg. | ASH | MSP | ReAct | SCALE | Avg. |
| AM | 82.16 | 77.49 | 79.94 | 83.61 | 80.8 | 59.14 | 64.45 | 62.82 | 57.2 | 60.9 |
| AM+DiffAug | 83.62 | 78.35 | 81.29 | 84.8 | **82.02** | 55.13 | 62.7 | 59.71 | 54.27 | **57.95** |
| RN50 | 78.17 | 76.02 | 77.38 | 81.36 | 78.23 | 63.32 | 65.68 | 66.69 | 59.79 | 63.87 |
| RN50+DiffAug | 79.86 | 76.86 | 78.76 | 82.8 | **79.57** | 60.21 | 64.91 | 62.84 | 56.15 | **61.03** |
| DAM | 74.16 | 75.2 | 75.14 | 77.07 | 75.39 | 66.34 | 67.42 | 67.72 | 63.67 | 66.29 |
| DAM+DiffAug | 75.73 | 75.65 | 75.87 | 78.56 | **76.45** | 64.99 | 66.56 | 66.28 | 61.94 | **64.94** |
| DA | 79.14 | 76.67 | 78.43 | 81.52 | 78.94 | 67.44 | 65.9 | 65.9 | 63.74 | 65.75 |
| DA+DiffAug | 79.54 | 76.92 | 79.1 | 81.42 | **79.25** | 66.52 | 65.41 | 64.25 | 63.19 | **64.84** |

Table 16: AUROC and FPR on SSB-Hard Dataset of ImageNet Near-OOD Detection.

| Train Augmentation | AUROC | | | | | FPR95 | | | | |
|---|---|---|---|---|---|---|---|---|---|---|
| | ASH | MSP | ReAct | SCALE | Avg. | ASH | MSP | ReAct | SCALE | Avg. |
| AM | 78.22 | 72.83 | 75.86 | 79.69 | 76.65 | 68.17 | 74.39 | 74.48 | 67.11 | 71.04 |
| AM+DiffAug | 80.48 | 73.81 | 77.2 | 81.7 | **78.3** | 63.31 | 72.88 | 72.43 | 63.25 | **67.97** |
| RN50 | 72.89 | 72.09 | 73.03 | 77.34 | 73.84 | 73.66 | 74.49 | 77.55 | 67.72 | 73.35 |
| RN50+DiffAug | 75.09 | 72.89 | 74.49 | 79.06 | **75.38** | 69.82 | 73.23 | 73.56 | 64.67 | **70.32** |
| DAM | 65.68 | 69.23 | 68.35 | 69.42 | 68.17 | 81.03 | 78.46 | 81.5 | 77.97 | 79.74 |
| DAM+DiffAug | 68.33 | 69.82 | 69.05 | 71.9 | **69.78** | 78.27 | 77.89 | 81.32 | 75.79 | **78.32** |
| DA | 76.65 | 72.35 | 75.28 | 78.59 | **75.72** | 72.26 | 75.27 | 75.27 | 70.4 | **73.3** |
| DA+DiffAug | 76.75 | 72.32 | 75.5 | 77.95 | 75.63 | 72.34 | 75.32 | 74.98 | 71.33 | 73.49 |

Table 17: AUROC and FPR on NINCO Dataset of ImageNet Near-OOD Detection.

| Train Augmentation | AUROC | | | | | FPR95 | | | | |
|---|---|---|---|---|---|---|---|---|---|---|
| | ASH | MSP | ReAct | SCALE | Avg. | ASH | MSP | ReAct | SCALE | Avg. |
| AM | 86.11 | 82.15 | 84.01 | 87.53 | 84.95 | 50.11 | 54.52 | 51.16 | 47.3 | 50.77 |
| AM+DiffAug | 86.75 | 82.88 | 85.39 | 87.91 | **85.73** | 46.95 | 52.52 | 46.98 | 45.3 | **47.94** |
| RN50 | 83.45 | 79.95 | 81.73 | 85.37 | 82.62 | 52.97 | 56.88 | 55.82 | 51.86 | 54.38 |
| RN50+DiffAug | 84.63 | 80.84 | 83.03 | 86.53 | **83.76** | 50.6 | 56.59 | 52.12 | 47.63 | **51.73** |
| DAM | 82.65 | 81.16 | 81.94 | 84.71 | 82.61 | 51.65 | 56.38 | 53.94 | 49.36 | 52.83 |
| DAM+DiffAug | 83.12 | 81.47 | 82.68 | 85.23 | **83.12** | 51.71 | 55.24 | 51.23 | 48.1 | **51.57** |
| DA | 81.62 | 80.99 | 81.58 | 84.45 | 82.16 | 62.62 | 56.52 | 56.52 | 57.07 | 58.18 |
| DA+DiffAug | 82.32 | 81.52 | 82.7 | 84.9 | **82.86** | 60.71 | 55.49 | 53.52 | 55.06 | **56.2** |

Table 18: ImageNet Near-OOD Detection results with ViT using MSP. Here, we observe comparable performance although we notice slight improvements in detection rates. We do not show the results for other OOD algorithms since we found those results to be significantly worse than simple MSP. This may be because the OOD research is mainly focused on deep convolution architectures such as DenseNet and ResNet.

| Train Augmentation | AUROC | | | FPR95 | | |
|---|---|---|---|---|---|---|
| | SSB-Hard | NINCO | Avg. | SSB-Hard | NINCO | Avg. |
| ViT | 72.02 | 81.61 | 76.82 | 82.19 | 61.87 | 72.03 |
| ViT+DiffAug | 72.22 | 82.00 | 77.11 | 81.42 | 58.06 | 69.74 |

## B.5 Ablation Experiments

We describe ablation experiments related to DiffAug and DiffAug Ensemble in the following.

### B.5.1 Extra Training

We train the pretrained AugMix checkpoint for extra 10 epochs to isolate the improvement obtained by DiffAug finetuning. In Table 19, we analyse the OOD detection performance as well as the robustness to covariate shift (Imagenet-C, Imagenet-R and Imagenet-S) finding no notable difference as compared to the results in Tables 3, 7 and 9.

### B.5.2 DiffAug Hyperparameters

The time range used to generate the DiffAug train augmentations constitutes the key hyperparameter and we analyse and compare between *weaker* DiffAug augmentations ($t \in [0, 500]$) and *stronger* DiffAug augmentations ($t \in [500, 999]$). From Table 19, we find that both weak and strong DiffAug augmentations complement each other and contribute to different aspects of robustness. Overall, using the entire diffusion time range to generate DiffAug yields consistent improvements.

Table 19: **Ablation Analysis**.

(a) Top-1 Accuracy(%) on ImageNet-C (severity=5) and ImageNet-Test. We observe that extra AugMix training does not introduce any remarkable difference with respect to the pretrained checkpoint allowing us to clearly attribute the improved robustness to DiffAug. Further, we also analyse the choice of diffusion time-range for generating the DiffAug augmentations and find that the stronger DiffAug augmentations generated with $t \in [500, 999]$ enhances robustness to Imagenet-C as compared to DiffAug $[0, 500]$ while obtaining slightly lower accuracy on Imagenet-Test in DE and DDA evaluation modes. Using the entire diffusion time-scale tends to achieve the right balance between both.

| Train Augmentations | ImageNet-C (severity = 5) | | | | | ImageNet-Test | | | | |
|---|---|---|---|---|---|---|---|---|---|---|
| | DDA | DDA (SE) | DE | Def. | **Avg** | DDA | DDA (SE) | DE | Def. | **Avg** |
| AM | 33.18 | 36.54 | 34.08 | 26.72 | 32.63 | 62.23 | 75.98 | 73.8 | 77.53 | 72.39 |
| AM+DiffAug | 34.64 | 38.61 | 38.58 | 29.47 | 35.33 | 63.53 | 76.09 | 75.88 | 77.34 | 73.21 |
| AM+Extra | 33.22 | 36.83 | 34.48 | 27.14 | 32.92 | 61.96 | 75.98 | 73.74 | 77.57 | 72.31 |
| AM+DiffAug[0,500] | 34.05 | 36.96 | 36.15 | 26.93 | 33.52 | 63.94 | 76.09 | 76.15 | 77.25 | **73.36** |
| AM+DiffAug[500,999] | 34.53 | 39.04 | 38.61 | 30.29 | **35.62** | 62.67 | 76.05 | 74.74 | 77.38 | 72.71 |

(b) Top-1 Accuracy(%) on ImageNet-R (severity=5) and ImageNet-S. As above, we find that extra Augmix training does not introduce any significant change. We also observe that DiffAug generated with [0,500] contributes more to improve robustness to Imagenet-R and Imagenet-S highlighting the benefits of using the entire diffusion time range for generating augmentations.

| Train Augmentations | ImageNet-R | | | | | ImageNet-S | | | | |
|---|---|---|---|---|---|---|---|---|---|---|
| | DDA | DDA (SE) | DE | Def. | **Avg** | DDA | DDA (SE) | DE | Def. | **Avg** |
| AM | 36.39 | 42.02 | 42.22 | 41.03 | 40.42 | 11.53 | 13.74 | 15.73 | 11.17 | 13.04 |
| AM+DiffAug | 37.09 | 42.14 | 43.28 | 40.98 | 40.87 | 12.30 | 14.02 | 16.00 | 10.95 | 13.32 |
| AM+Extra | 36.05 | 41.66 | 41.84 | 40.70 | 40.06 | 11.40 | 13.56 | 15.46 | 10.91 | 12.83 |
| AM+DiffAug[0,500] | 37.61 | 42.36 | 43.77 | 41.21 | **41.24** | 12.55 | 14.00 | 15.79 | 10.77 | **13.28** |
| AM+DiffAug[500,999] | 35.94 | 40.91 | 41.51 | 40.11 | 39.62 | 11.49 | 13.29 | 15.48 | 10.62 | 12.72 |

(c) OOD Detection. As above, we find that extra Augmix training does not introduce any significant change. Here, we find that DiffAug[500,999] contributes more to the improved OOD detection.

| TrainAugmentation | ASH | MSP | ReAct | Scale | **Avg.** |
|---|---|---|---|---|---|
| AM | 59.14 | 64.45 | 62.82 | 57.2 | 60.9 |
| AM+DiffAug | 55.13 | 62.7 | 59.71 | 54.27 | 57.95 |
| AM+Extra | 58.17 | 64.30 | 62.77 | 56.57 | 60.45 |
| AM+DiffAug[0,500] | 56.63 | 63.39 | 61.42 | 54.83 | 59.07 |
| AM+DiffAug[500,999] | 54.48 | 62.14 | 58.63 | 53.63 | **57.22** |

### B.5.3 DiffAug Ensemble (DE) Hyperparameters

We use set $S = \{0, 50, \cdots, 450\}$ to compute the DE accuracy in Table 9. Here, we study the effect of step-size (the difference between consecutive times) and the maximum diffusion time used. First, we analyse the performance when using maximum diffusion time $= 999$ instead of $t = 450$ in Fig. 13. Then, using maximum diffusion time $= 450$, we study the effect of using alternative step-sizes of 25 (more augmentations) and 75 (fewer augmentations) in Fig. 12.

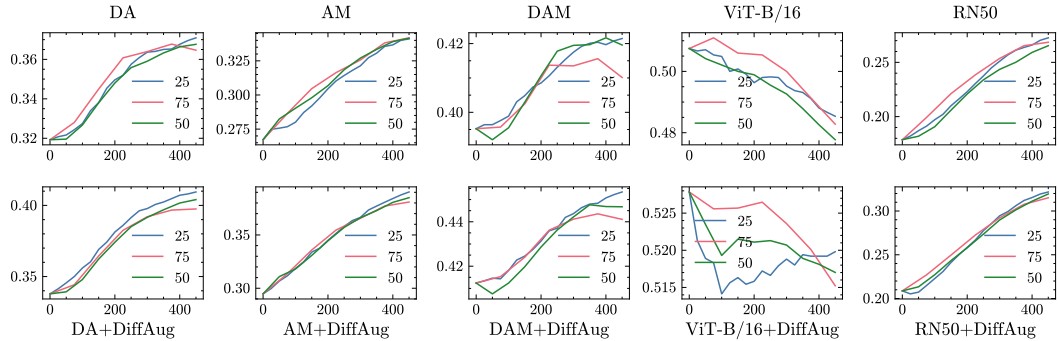

Figure 12: Plots of $t$ vs DE Accuracy on Imagenet-C (severity=5) for different step-sizes: in general, we observe that the performance is largely robust to the choice of step-size although using $t = 25$ gives slightly improved result.

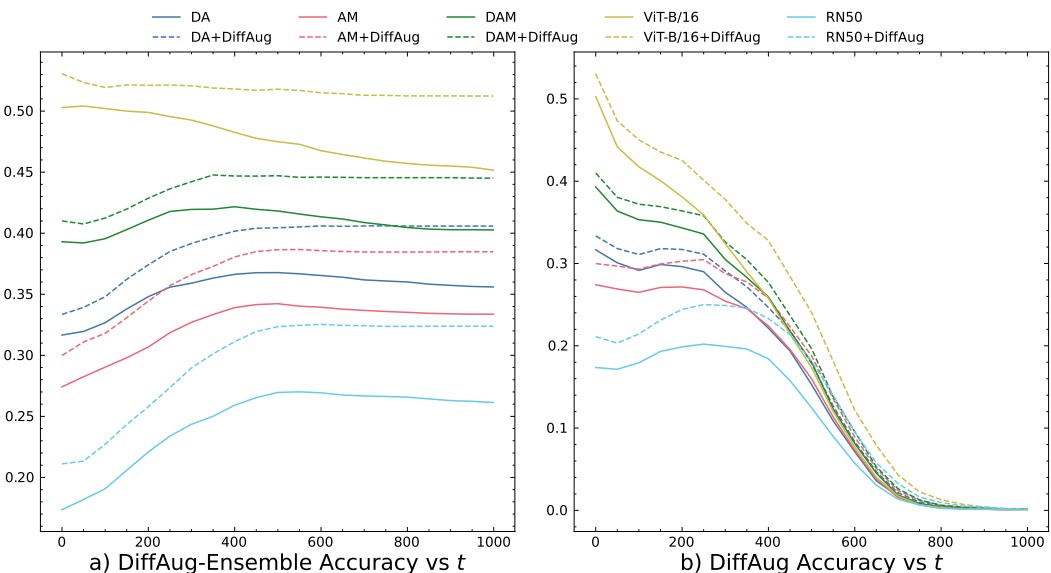

Figure 13: Plots of $t$ vs DE Accuracy and DiffAug Accuracy on Imagenet-C (severity=5): in general, we observe that the performance saturates beyond a certain time-step although the corresponding DiffAug accuracy steadily decreases. These plots also highlight the robustness of straightforward averaging as a test-time augmentation method.

### B.5.4 Conditional DiffAug

In this section, we study the application of DiffAug with conditioning. For these experiments, we utilize two — one class-conditional and one unconditional — ImageNet256 diffusion models to apply the DiffAug augmentation. If $s_{\theta_1}(\mathbf{x}|y)$ and $s_{\theta_2}(\mathbf{x})$ refer to the conditional and unconditional scores, we can combine these using an hyperparameter $\lambda$ (similar to classifier-free guidance) and use the following formulation of $s_\theta$ to apply the DiffAug augmentation (Eq. (3)):

$$s_\theta(\mathbf{x}, t) = s_{\theta_1}(\mathbf{x}, t) + \lambda(s_{\theta_2}(\mathbf{x}|y, t) - s_{\theta_1}(\mathbf{x}, t)) \tag{9}$$

where, $\lambda = 0$ denotes unconditional DiffAug, as explored in the main paper. In the following, we illustrate conditional DiffAug both qualitatively and quantitatively.

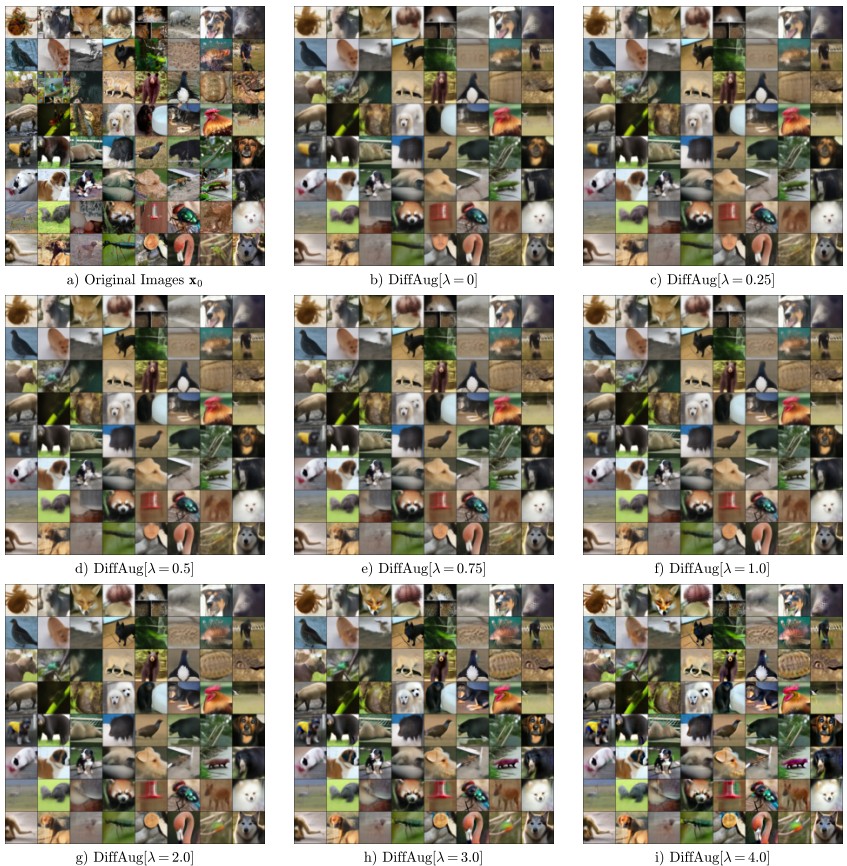

a) Original Images $\mathbf{x}_0$     b) DiffAug[$\lambda = 0$]     c) DiffAug[$\lambda = 0.25$]

d) DiffAug[$\lambda = 0.5$]     e) DiffAug[$\lambda = 0.75$]     f) DiffAug[$\lambda = 1.0$]

g) DiffAug[$\lambda = 2.0$]     h) DiffAug[$\lambda = 3.0$]     i) DiffAug[$\lambda = 4.0$]

Figure 14: We illustrate the DiffAug augmentations for various values of $\lambda$ at $t = 600$.

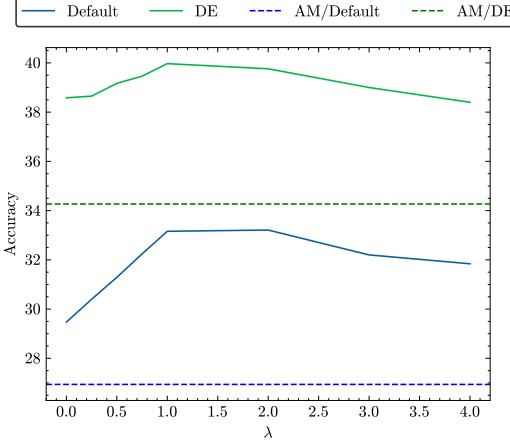

Figure 15: We extend AugMix(AM) with DiffAug using different values of $\lambda$ and plot the ImageNet-C (severity=5) accuracy for both default and DE inference. We observe that conditional DiffAug can enhance performance for optimal values of $\lambda$. Nevertheless, DiffAug can also be applied with unconditional diffusion models broadening its applications.

### B.5.5 DiffAug with DiT

In this section, we explore the application of DiffAug with latent-space diffusion models such as the Diffusion-Transformer (DiT). Diffusion-Transformer models the distribution of a variational autoencoder (VAE) latent-space. To apply DiffAug with DiT, we (i) first apply the VAE encoder to encode the train image into the latent-space, (ii) apply DiffAug within the latent-space, and (iii) finally, decode the image using the VAE decoder. We refer to this as DiT-DiffAug. We finetune the torchvision ResNet-50 model for 10 epochs using class-conditional and unconditional DiT-DiffAug augmentation and summarize the results in Table 20.

Table 20: In this table, we explore DiffAug augmentation with DiT, a latent-space diffusion model. While the test-accuracy is preserved even when training upon partially synthesized DiT images, we observe a smaller robustness improvement as compared to the DiffAug with pixel-space models.

| Method | ImageNet-Test | ImageNet-C |
|---|---|---|
| RN50 | 76.15 | 17.87 |
| +DiffAug | 75.95 | 20.87 |
| +DiT-DiffAug | 76.08 | 18.91 |
| +DiT-DiffAug [Conditional] | 76.09 | 19.27 |

### B.6 Derivation of Theorem 4.1

For the forward-diffusion SDEs considered in this paper, the marginal distribution $p_t(\mathbf{x})$ can be expressed in terms of the data distribution $p(\mathbf{x}_0)$:

$$p_t(\mathbf{x}) = \int_{\mathbf{x}_0} p_t(\mathbf{x}|\mathbf{x}_0)p(\mathbf{x}_0)d\mathbf{x}_0 \tag{10}$$

where

$$p_t(\mathbf{x}|\mathbf{x}_0) = \mathcal{N}(\mathbf{x} \mid \mu(\mathbf{x}_0, t), \ \sigma^2(t)\mathbf{I}).$$

If we denote $\mu(\mathbf{x}_0, t)$ by $\mathbf{m}_t$, we can rewrite $p_t(\mathbf{x})$ as

$$p_t(\mathbf{x}) = \int_{\mathbf{m}_t} p_t(\mathbf{x}|\mathbf{m}_t)p(\mathbf{m}_t)d\mathbf{m}_t$$

since $\mu$ is linear and invertible. The optimal score-function $s_{\theta*}(\mathbf{x}, t) = \nabla_{\mathbf{x}} \log p_t(\mathbf{x})$ can be simplified as:

$$s_{\theta*}(\mathbf{x}, t) = \frac{1}{p_t(\mathbf{x})} \int_{\mathbf{m}_t} \frac{\mathbf{m}_t - \mathbf{x}}{\sigma^2(t)} \, p_t(\mathbf{x}|\mathbf{m}_t) \, p(\mathbf{m}_t)d\mathbf{m}_t \tag{11}$$

Using Eq (11), we can rewrite the denoised example, $\hat{\mathbf{x}}_t = \mathbf{x} + \sigma^2(t)s_\theta(\mathbf{x}, t)$, as:

$$\hat{\mathbf{x}}_t = \frac{1}{p_t(\mathbf{x})} \int_{\mathbf{m}_t} \mathbf{m}_t \, p_t(\mathbf{x}|\mathbf{m}_t) \, p(\mathbf{m}_t)d\mathbf{m}_t = \int_{\mathbf{m}_t} \mathbf{m}_t \, p_t(\mathbf{m}_t|\mathbf{x})d\mathbf{m}_t = \mathbb{E}[\mathbf{m}_t|\mathbf{x}] \tag{12}$$

That is, the denoised example $\hat{\mathbf{x}}_t$ is in fact the expected value of the mean $\mathbf{m}_t$ given input $\mathbf{x}$. (See also Eq (4) in the main text ).

To compute $\frac{\partial \hat{\mathbf{x}}_t}{\partial \mathbf{x}}$, we algebraically simplify $\int_{\mathbf{m}_t} \mathbf{m}_t \nabla_{\mathbf{x}} \left( \frac{p_t(\mathbf{x}|\mathbf{m}_t)}{p_t(\mathbf{x})} \right) p(\mathbf{m}_t) d\mathbf{m}_t$ as follows:

$$
\begin{aligned}
\frac{\partial \hat{\mathbf{x}}_t}{\partial \mathbf{x}} &= \int_{\mathbf{m}_t} \mathbf{m}_t \left( \frac{\nabla_{\mathbf{x}} p_t(\mathbf{x}|\mathbf{m}_t)}{p_t(\mathbf{x})} - \frac{p_t(\mathbf{x}|\mathbf{m}_t) \nabla_{\mathbf{x}} p_t(\mathbf{x})}{p_t^2(\mathbf{x})} \right)^\top p(\mathbf{m}_t) d\mathbf{m}_t \\
&= \int_{\mathbf{m}_t} \mathbf{m}_t \left( \frac{p_t(\mathbf{x}|\mathbf{m}_t)}{p_t(\mathbf{x})} \frac{\mathbf{m}_t - \mathbf{x}}{\sigma^2(t)} - \frac{p_t(\mathbf{x}|\mathbf{m}_t) \nabla_{\mathbf{x}} \log p_t(\mathbf{x})}{p_t(\mathbf{x})} \right)^\top p(\mathbf{m}_t) d\mathbf{m}_t \\
&= \int_{\mathbf{m}_t} \mathbf{m}_t \frac{p_t(\mathbf{x}|\mathbf{m}_t)}{p_t(\mathbf{x})} \left( \frac{\mathbf{m}_t - \mathbf{x}}{\sigma^2(t)} - \nabla_{\mathbf{x}} \log p_t(\mathbf{x}) \right)^\top p(\mathbf{m}_t) d\mathbf{m}_t \\
&= \int_{\mathbf{m}_t} \mathbf{m}_t \frac{p_t(\mathbf{x}|\mathbf{m}_t)}{p_t(\mathbf{x})} \left( \frac{\mathbf{m}_t - \mathbf{x} - \sigma^2(t) \nabla_{\mathbf{x}} \log p_t(\mathbf{x})}{\sigma^2(t)} \right)^\top p(\mathbf{m}_t) d\mathbf{m}_t \\
&= \int_{\mathbf{m}_t} \mathbf{m}_t \frac{p_t(\mathbf{x}|\mathbf{m}_t)}{p_t(\mathbf{x})} \left( \frac{\mathbf{m}_t}{\sigma^2(t)} - \frac{\mathbf{x} + \sigma^2(t) \nabla_{\mathbf{x}} \log p_t(\mathbf{x})}{\sigma^2(t)} \right)^\top p(\mathbf{m}_t) d\mathbf{m}_t \\
&= \int_{\mathbf{m}_t} \frac{\mathbf{m}_t \mathbf{m}_t^\top}{\sigma^2(t)} p_t(\mathbf{m}_t|\mathbf{x}) d\mathbf{m}_t - \left( \int_{\mathbf{m}_t} \mathbf{m}_t \, p_t(\mathbf{m}_t|\mathbf{x}) d\mathbf{m}_t \right) \frac{\hat{\mathbf{x}}_t^\top}{\sigma^2(t)} \\
&= \frac{1}{\sigma^2(t)} \left( \int_{\mathbf{m}_t} \mathbf{m}_t \mathbf{m}_t^\top p_t(\mathbf{m}_t|\mathbf{x}) d\mathbf{m}_t - \hat{\mathbf{x}}_t \hat{\mathbf{x}}_t^\top \right) \\
&= \frac{1}{\sigma^2(t)} \left( \mathbb{E}[\mathbf{m}_t \mathbf{m}_t^\top | \mathbf{x}] - \mathbb{E}[\mathbf{m}_t | \mathbf{x}] \mathbb{E}[\mathbf{m}_t | \mathbf{x}]^\top \right) \\
&= \frac{1}{\sigma^2(t)} \mathrm{Cov}[\mathbf{m}_t | \mathbf{x}]
\end{aligned}
\tag{13}
$$

## B.7 Analysis with SVD Decomposition

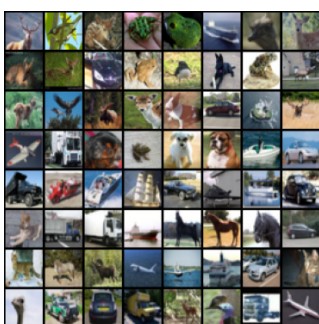

Figure 16: CIFAR10 examples used for SVD Analysis

Considering the CIFAR10 images in Fig. 16, we compute the $3072 \times 3072$ jacobian matrix $J$ and then apply SVD decomposition $J = USV$ using default settings in PyTorch. Then, we visualise — after min-max normalization — the columns of $U$ and rows of $V$ along with the (batch-averaged) value in the diagonal matrix $S$ of the corresponding row/column. We find that the principal components of the jacobian matrix are perceptually aligned and provides additional intuition for Theorem 4.1. See Figs. 17 and 18.

## C  Appendix for Section 5

### C.1  Guidance Classifier Training: Experiment Details

We use the pretrained noisy Imagenet classifier released by Dhariwal and Nichol [14] while we trained the noisy CIFAR10 classifier ourselves; the Imagenet classifier is the downsampling half of the UNET with attention pooling classifier-head while we use WideResNet-28-2 as the architecture for CIFAR10. For the DA-classifier, we simply add an extra convolution that can process the

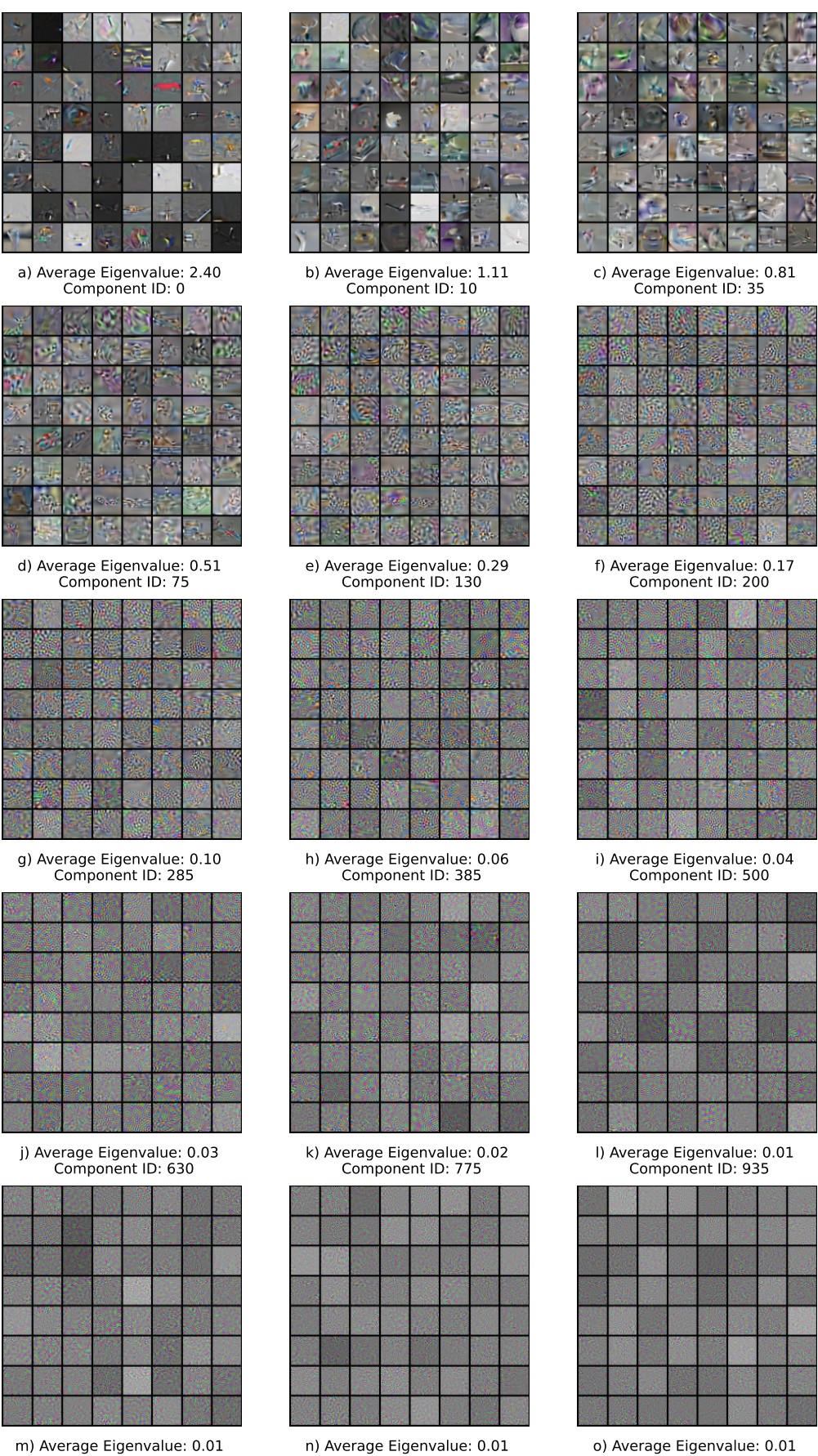

a) Average Eigenvalue: 2.40
Component ID: 0

b) Average Eigenvalue: 1.11
Component ID: 10

c) Average Eigenvalue: 0.81
Component ID: 35

d) Average Eigenvalue: 0.51
Component ID: 75

e) Average Eigenvalue: 0.29
Component ID: 130

f) Average Eigenvalue: 0.17
Component ID: 200

g) Average Eigenvalue: 0.10
Component ID: 285

h) Average Eigenvalue: 0.06
Component ID: 385

i) Average Eigenvalue: 0.04
Component ID: 500

j) Average Eigenvalue: 0.03
Component ID: 630

k) Average Eigenvalue: 0.02
Component ID: 775

l) Average Eigenvalue: 0.01
Component ID: 935

m) Average Eigenvalue: 0.01
Component ID: 1110

n) Average Eigenvalue: 0.01
Component ID: 1300

o) Average Eigenvalue: 0.01
Component ID: 1505

Figure 17: Columns of $U$

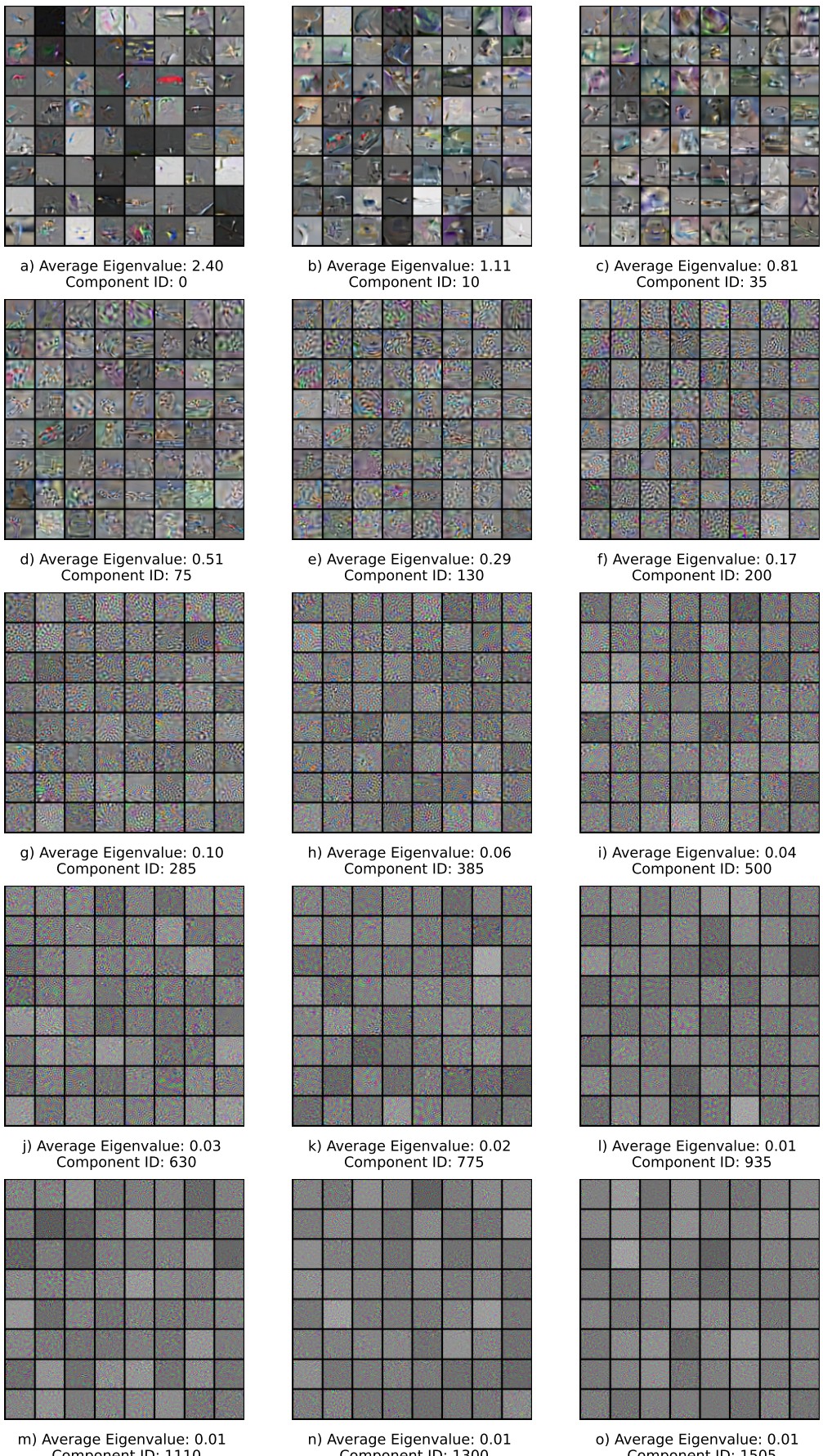

a) Average Eigenvalue: 2.40
Component ID: 0

b) Average Eigenvalue: 1.11
Component ID: 10

c) Average Eigenvalue: 0.81
Component ID: 35

d) Average Eigenvalue: 0.51
Component ID: 75

e) Average Eigenvalue: 0.29
Component ID: 130

f) Average Eigenvalue: 0.17
Component ID: 200

g) Average Eigenvalue: 0.10
Component ID: 285

h) Average Eigenvalue: 0.06
Component ID: 385

i) Average Eigenvalue: 0.04
Component ID: 500

j) Average Eigenvalue: 0.03
Component ID: 630

k) Average Eigenvalue: 0.02
Component ID: 775

l) Average Eigenvalue: 0.01
Component ID: 935

m) Average Eigenvalue: 0.01
Component ID: 1110

n) Average Eigenvalue: 0.01
Component ID: 1300

o) Average Eigenvalue: 0.01
Component ID: 1505

Figure 18: Rows of $V$

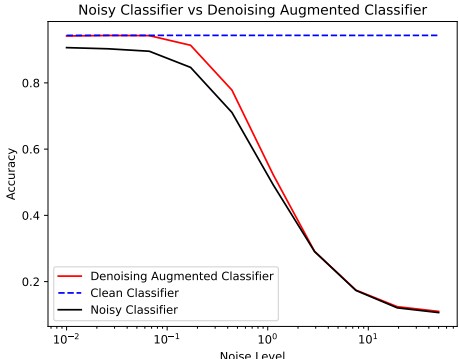

Figure 19: CIFAR10: Test Accuracy vs. Noise Scale.

denoised input: for Imagenet, we finetune the pretrained noisy classifier by adding an additional input-convolution module while we train the denoising-augmented CIFAR10 classifier from scratch. The details of the optimization are as follows: (1) for Imagenet, we fine-tune the entire network along with the new convolution-module (initialized with very small weights) using AdamW optimizer with a learning-rate of 1e-5 and a weight-decay of 0.05 for 50k steps with a batch size of 128. (2) For CIFAR10, we train both noisy and DA-classifiers for 150k steps with a batch size of 512 using AdamW optimizer with a learning-rate of 3e-4 and weight decay of 0.05. For CIFAR10 classifiers, we use the Exponential Moving Average of the parameters with decay-rate equal to 0.999.

## C.2 Classifier-Guided Diffusion: Sampling

We use a PC sampler as described in Song et al. [47] with 1000 discretization steps for CIFAR10 samples while we use a DDIM [46] sampler with 25 discretization steps for Imagenet samples. We use the 256x256 class-conditional diffusion model open-sourced by Dhariwal and Nichol [13] for our Imagenet experiments and set the classifier scale $\lambda_s = 2.5$ following their experimental setup for DDIM-25 samples. The classifier-scale is set to 1.0 for CIFAR10 experiments.

## C.3 Comparisons with DLSM, ECT and Robust Guidance

Table 21: DLSM vs DA-Classifier: In this table, we compare between using DLSM – i.e., DLSM-Loss in addition to cross-entropy loss in training classifiers on noisy images as input – and DA-Classifiers wherein we use both noisy and denoised images as input but only used cross-entropy loss for training. Since Chao et al. [6] use ResNet18 backbones for their CIFAR10 experiments, we train a separate DA-Classifier for these comparisons. We compare between FID, IS and also compare the unconditional precision and recall (P/R) and the average class-conditional $\widetilde{\text{Precision}}/\widetilde{\text{Recall}}/\widetilde{\text{Density}}/\widetilde{\text{Coverage}}$.We obtain our results for Noisy Classifier (CE) and Noisy Classifier (DLSM) from Table 2 of Chao et al. [6]. While the FID and IS scores are comparable, we note that our class-wise Precision, Recall, Density and Coverage metrics are either comparable or demonstrate a significant improvement.

| Method | FID↓ | IS ↑ | P ↑ | R ↑ | $\widetilde{P}$ ↑ | $\widetilde{R}$ ↑ | $\widetilde{D}$ ↑ | $\widetilde{C}$ ↑ |
|---|---|---|---|---|---|---|---|---|
| Noisy Classifier (CE) | 4.10 | 9.08 | 0.67 | 0.61 | 0.51 | 0.59 | 0.63 | 0.60 |
| Noisy Classifier (DLSM) | **2.25** | 9.90 | 0.65 | 0.62 | 0.56 | 0.61 | 0.76 | 0.71 |
| DA-Classifier | 2.27 | **9.91** | 0.64 | 0.62 | **0.63** | **0.64** | **0.90** | **0.77** |

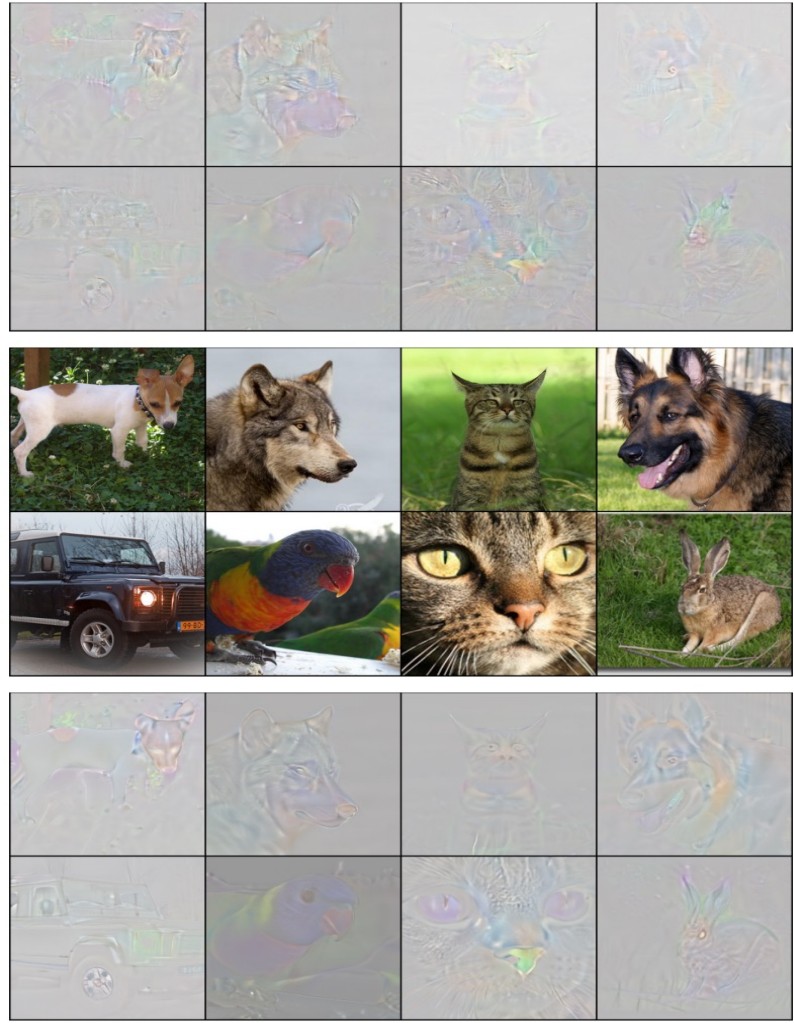

Figure 20: Min-max normalized gradients on samples diffused to $t = 300$ ($T = 999$). Top panel: gradients obtained with noisy classifier. Bottom panel: gradients obtained with DA-classifier. Middle panel: corresponding clean Imagenet samples. We recommend zooming in to see differences between gradients, e.g. the clearer coherence in DA-classifier gradients.

Table 22: ED and Robust-Guidance vs DA-Classifier: Zheng et al. [58] propose two complementary techniques to improve over vanilla classifier-guidance: Entropy-Constraint Training (ECT) and Entropy-Driven Sampling (EDS). ECT consists of adding an additional loss term to the cross-entropy loss encouraging the predictions to be closer to uniform distribution (similar to the label-smoothing loss). EDS modifies the sampling to use a diffusion-time dependent scaling factor designed to address premature vanishing guidance-gradients. The sampling method (EDS/Vanilla) can be chosen independent of the training method (determined by the loss-objective and classifier-inputs). In the following, we compare between ECT and DA-Classifiers using Vanilla Sampling method using the results in Table 3 of Zheng et al. [58]. As the robust-classifier [29] was not evaluated for Imagenet-256, we fine-tuned the open-source checkpoint using the open-source code provided by robust-guidance for 50k steps with learning rate=1e-5 We observe that DA-Classifier obtains better FID/IS than both ECT and Robust-Guidance.

| Method | Loss-Objective | Classifier-Inputs | FID | sFID | IS | P | R |
|---|---|---|---|---|---|---|---|
| Noisy-Classifier | CE | Noisy Image | 5.46 | 5.32 | 194.48 | 0.81 | 0.49 |
| ECT-Classifier | CE+ECT | Noisy Image | 5.34 | **5.3** | 196.8 | 0.81 | 0.49 |
| Robust-Classifier | CE + Adv. Training | Noisy Image | 5.44 | 5.81 | 142.61 | 0.74 | **0.56** |
| DA-Classifier | CE | Noisy Image & Denoised Image | **5.24** | 5.37 | **201.72** | 0.81 | 0.49 |

$$\frac{d\log p_\phi}{d\mathbf{x}} \qquad \frac{\partial \log p_\phi}{\partial \mathbf{x}} \qquad \frac{\partial \log p_\phi}{\partial \hat{\mathbf{x}}} \qquad \frac{\partial \log p_\phi}{\partial \hat{\mathbf{x}}}\frac{\partial \hat{\mathbf{x}}}{\partial \mathbf{x}}$$

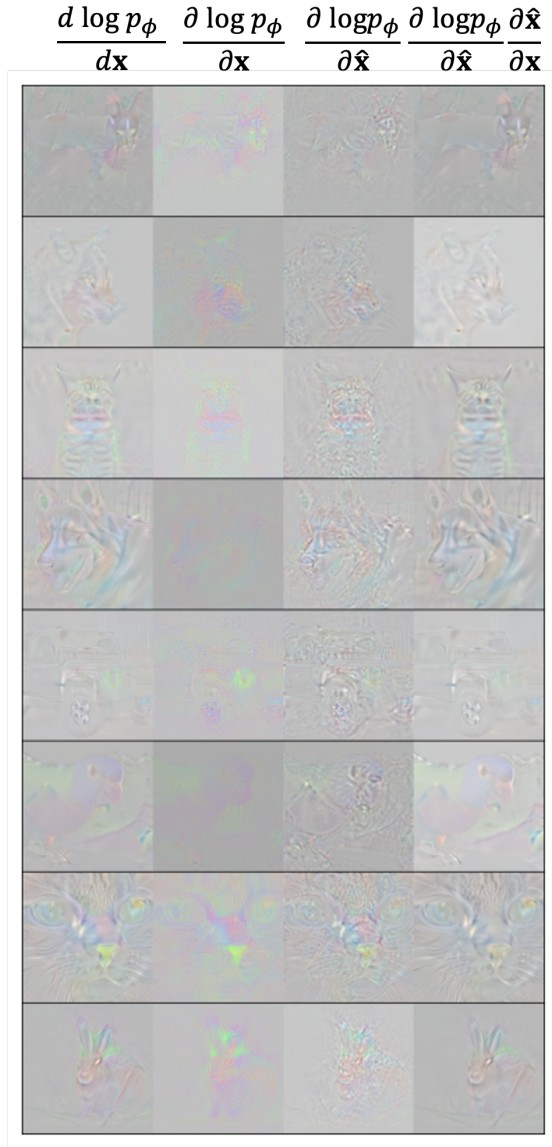

Figure 21: The figure shows the total derivative $\frac{d\log p_\phi(y|\mathbf{x},\hat{\mathbf{x}}_t,t)}{d\mathbf{x}} = \frac{\partial \log p_\phi}{\partial \mathbf{x}} + \frac{\partial \log p_\phi}{\partial \hat{\mathbf{x}}_t}\frac{\partial \hat{\mathbf{x}}_t}{\partial \mathbf{x}}$, the partial derivative with respect to noisy input $\frac{\partial \log p_\phi}{\partial \mathbf{x}}$, the partial derivative with respect to denoised input $\frac{\partial \log p_\phi}{\partial \hat{\mathbf{x}}}$, and $\frac{\partial \log p_\phi}{\partial \hat{\mathbf{x}}}\frac{\partial \hat{\mathbf{x}}_t}{\partial \mathbf{x}}$.

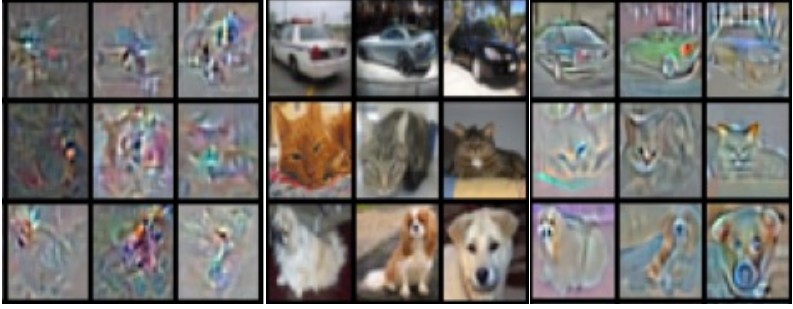

Figure 22: Min-max normalized gradients on samples diffused to $t = 0.35$ ($T = 1.0$). Left panel: gradients obtained with noisy classifier. Right panel: gradients obtained with the DA-classifier. Middle panel: clean corresponding CIFAR10 samples.

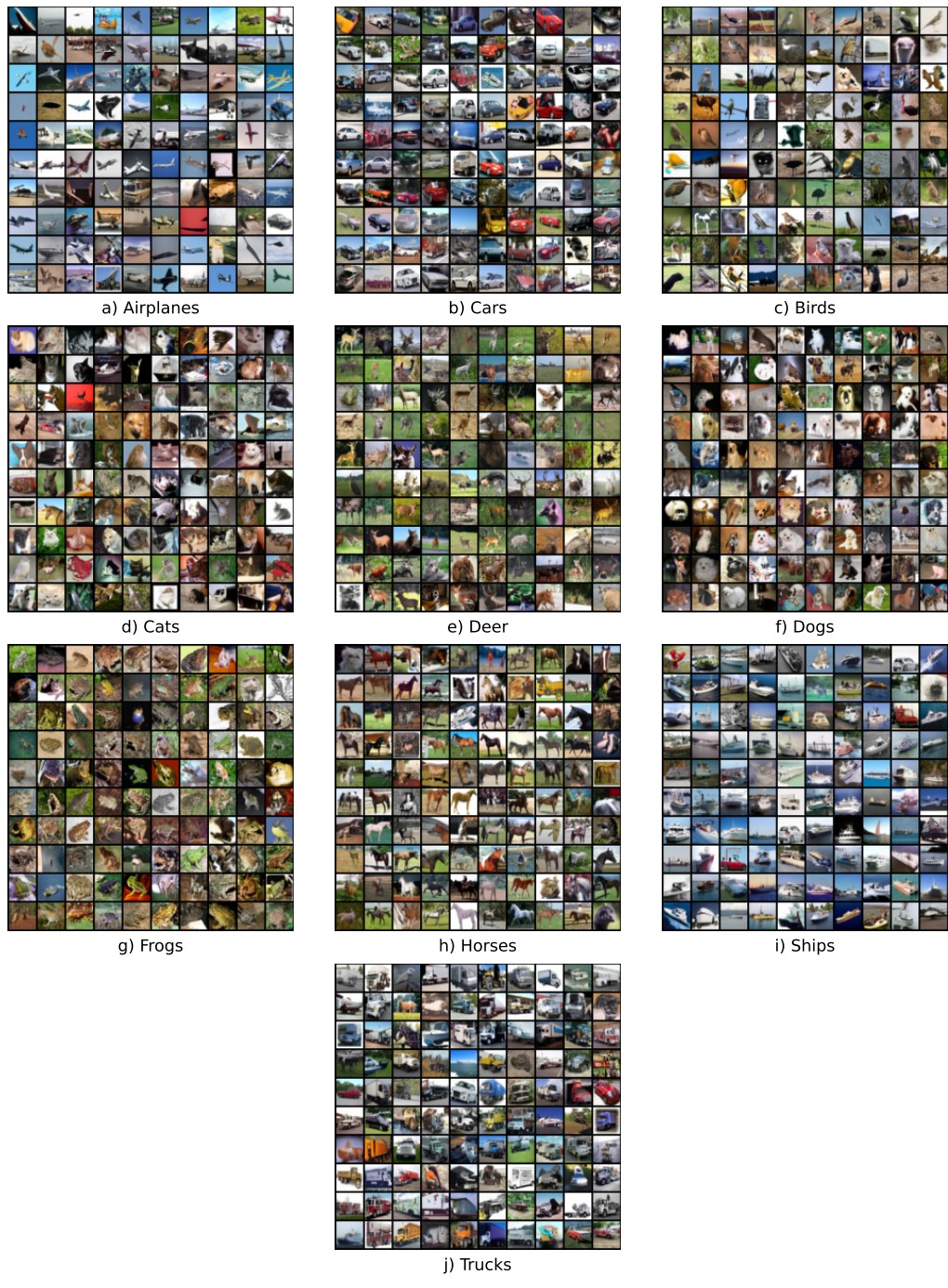

Figure 23: Uncurated CIFAR10 Samples with Noisy-Classifier.

## D   Uncurated Samples

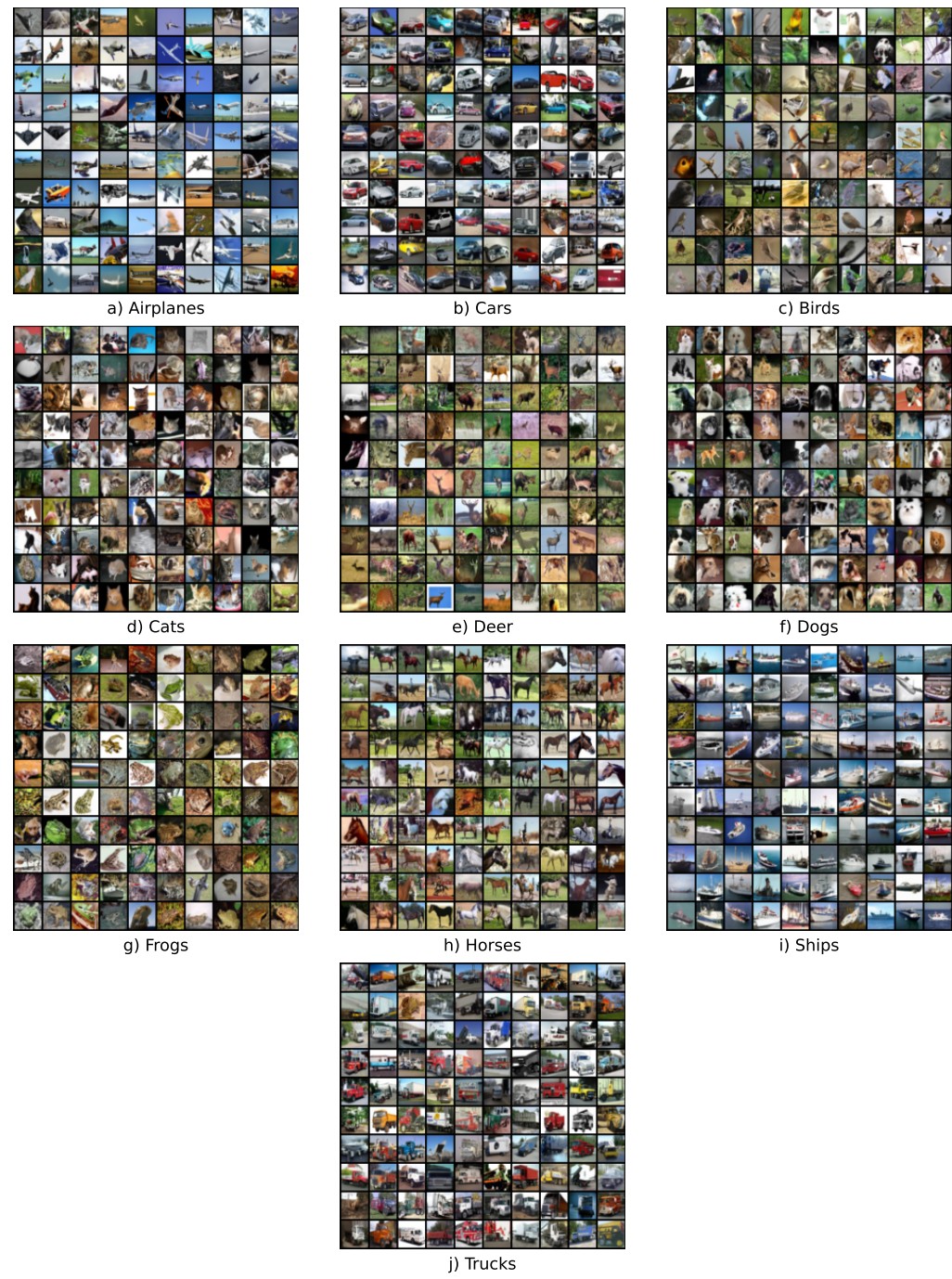

Figure 24: Uncurated CIFAR10 Samples with DA-classifier.

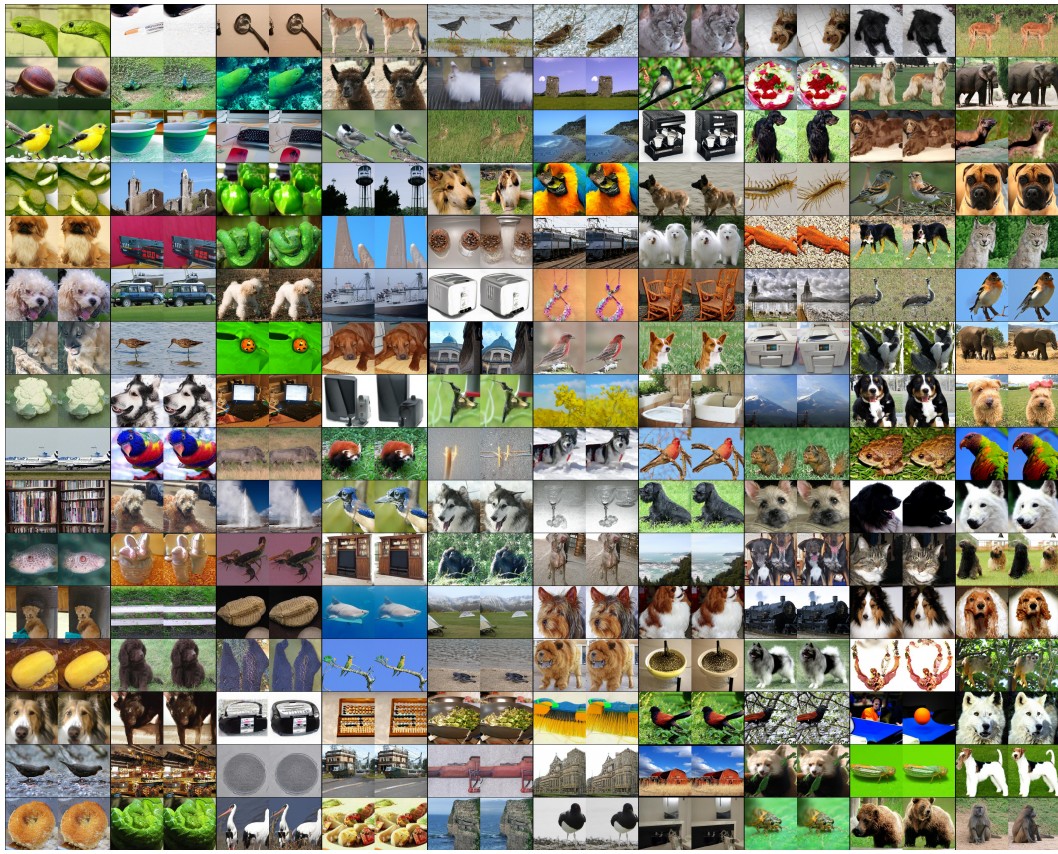

Figure 25: Uncurated generated samples (with images containing human faces removed) to compare between Noisy classifier (left) and DA-Classifier (right). Please zoom in to see the subtle improvements introduced by DA-Classifier guidance.

