# OpenReview forum: "DiffAug: A Diffuse-and-Denoise Augmentation for Training Robust Classifiers"
_NeurIPS.cc/2024/Conference — NeurIPS 2024 poster_

### Official Review · Reviewer_QY34 · 2024-06-25

**Soundness:** 3
**Presentation:** 3
**Contribution:** 3
**Rating:** 7
**Confidence:** 3

**Summary:**

The paper proposes DiffAug, a new method for training image classifiers that are more robust. DiffAug is based on diffusion models and effective at improving classifier robustness in several ways, such as resistance to variations in the data. It also can improve the performance of classifier-guided diffusion models. Furthermore, DiffAug is computationally efficient and can be combined with other augmentation techniques.

**Strengths:**

1. The writing and the presentation of the work is good and easy to follow.
2. The finding, that training with degraded images from the diffusion process won’t harm the classifier’s performance but even improve it, is interesting.
3. The experiments are comprehensive and diverse, showing the effectiveness of the proposed method.

**Weaknesses:**

While I currently find the paper favorable and have no major weaknesses to point out, I do have some questions that the authors' responses could help clarify (see the part below). I am open to raising the score based on their explanations.

**Questions:**

1.	The authors state on Line 106 that they are unaware of prior studies training classifiers with denoised examples. However, Line 20 mentions previous work using "synthetic images generated using Imagen [38]". Does reference [38] not qualify as a "previous study on training classifiers with denoised examples"? Perhaps the authors meant there's no prior work on training with diffused-and-denoised data?
2.	The authors mention "include Eq. 5 as an additional optimization objective" (Line 128-129). Could they elaborate on why it's considered "additional"? How does Eq. 5 differ from the standard classification loss typically used during classifier training?
3.	In Section 4, the experiments explore "unconditional improved-DDPM." Have the authors investigated using DiffAug with "conditional" diffusion models? Can such text guidance improve DiffAug's performance, or would the "conditional" process limit augmentation diversity and hinder performance?
4.	In Table 1, which model structure is trained on when evaluating “AM, DA, and DAM”? It seems the first column didn’t present enough information on that.
5.	Table 1 results (columns 3, 4 & 8, 9) consistently show DE underperforming compared to DDA(SE). What explains this difference? Could it be due to DDA's "self-ensemble" strategy (although DE also uses ensembles)? Does this suggest that using even multiple-step-denoised examples can enhance classifier robustness?
6.	While the paper focuses on improving classifier’s robustness based on diffusion models, there's a complementary area of research: diffusion-based attacks (DiffAttack [1], Diff-PGD [2]). How effective is DiffAug against these attacks? Given Diff-PGD's constraints with perturbation norms, the results in the paper on certified adversarial accuracy might generalize. However, DiffAttack is unrestricted. Can DiffAug still be applied in such cases? The authors should supplement these discussions in the paper for better evidence of DiffAug’s robustness.
7.	About the related work, the idea of leveraging a diffusion model for augmentation and incorporating denoised examples to improve classifier robustness was also mentioned in the discussion section of DiffAttack [1]. Citing this related work would be appropriate.
8.	Both DE and DDS employ a single reverse diffusion step during testing. What are the key distinctions between these methods? There seem no comparisons between them or descriptions about their differences.

[1] Chen, Jianqi, et al. "Diffusion models for imperceptible and transferable adversarial attack." arXiv preprint arXiv:2305.08192 (2023).

[2] Xue, Haotian, et al. "Diffusion-Based Adversarial Sample Generation for Improved Stealthiness and Controllability." arXiv preprint arXiv:2305.16494 (2023).

**Limitations:**

Yes. Limitations and social impacts have been involved in the paper.

---

> ### Author Rebuttal · Authors · 2024-08-07
>
> We thank you for your insightful reviews and affirmative evaluation of our work. We thank you for your appreciation of our presentation and method’s simplicity/computational efficiency. We agree with you that it is indeed interesting to learn that classifiers can be improved without sacrificing test accuracy through the use of degraded images from the diffusion process! We answer your questions in the following:
>
> **[Q1]** Thank you for this question. The reviewer is correct that we meant no prior work on training with diffused-and-denoised data, and in particular by “denoised” we meant “partially denoised”; we apologize for this confusion, and we will make sure to clarify this. In [38], the generated synthetic images are of high-quality since they use iterative denoising: in fact, to ensure better quality, they first fine-tune the Imagen model on ImageNet data and then use FID to select the model checkpoints and various hyperparameters of the sampling algorithm.
> ***
> **[Q2]** Additional optimization objective refers to extra loss term. When extending previous methods with DiffAug, we modify their official code by simply adding $\mathcal{L}$ in Eq. 5 to the original loss-terms. For example, AugMix involves 2 loss-terms: a cross-entropy loss term and a Jensen-Shannon divergence loss term. When extending AugMix with DiffAug, we introduce Eq.5 as the third loss-term.
> ***
> **[Q3]**  Similar to many standard augmentation techniques, we do not rely upon labels for DiffAug broadening its applicability for different tasks and scenarios. For example, DiffAug can be applied at both train-time and test-time (when class labels are unknown). Furthermore, it enables direct use of unlabeled data or a mix of labelled and unlabeled data to train the diffusion model. In theory, it should be possible to achieve performance enhancements with DiffAug by using conditional diffusion models, especially with innovative choices of prompt. However, this introduces additional hyperparameters such as the guidance-strength and choice of prompt. Additionally, it requires two forward passes through the diffusion model (one unconditional and one conditional) per training step. The augmentation diversity may certainly be affected especially if the guidance-strength is very high: in such cases, it is also possible that the classifier could cheat by exploiting imperceptible image statistics (e.g., due to adaptive layer-norms).
> ***
> **[Q4]** We use the official pretrained checkpoints for these models and all of them are trained using a ResNet-50 backbone. We will clarify this.
> ***
> **[Q5]** As compared to DE, DDA is a more complex technique that utilises multiple-diffusion steps to transform a test-sample from an unknown distribution into the source distribution. We find that the multiple-step denoising can be beneficial when dealing with severe corruptions in ImageNet-C. For example, we find negligible difference between DDA and DE over uncorrupted ImageNet test examples when considering the DiffAug-trained models (76.60 & 76.67 respectively). Yes, an optimal multiple-step denoising approach that effectively transforms the image into the source distribution may further enhance classifier robustness on ImageNet-C. For other distribution shifts such as ImageNet-R and ImageNet-S, we observe that DE offers better test-time adaptation on average as compared to DDA.
> ***
> **[Q6,7]**  We thank you for sharing these works on novel methods for diffusion-based adversarial example generation. We agree with your analysis that DiffAug’s certified adversarial accuracy results may generalise to Diff-PGD. From our understanding of DiffAttack, it produces transferable untargeted adversarial attacks at a high success rate and hence, we will need to conduct an empirical analysis to study DiffAug’s effectiveness against these examples. While we are interested in understanding the performance of DiffAug and DiffAug-Ensemble against DiffAttack, we were unable to perform this experiment during the rebuttal period due to time and resource constraints. We will include a discussion on these methods and aim to provide a preliminary empirical analysis in the final submission for completeness.
>
> In the following, we describe additional results on ImageNet-D, another stable-diffusion generated dataset designed to evaluate robustness (similar to DiffAttack). From Table 2 (global PDF), we find that extra synthetic data (from stable-diffusion, as described in global response) offers the most improvements against ImageNet-D --– similar to the suggestion in the DiffAttack paper. Interestingly, DiffAug training and DiffAug-Ensemble (DE) inference can offer further improvements: for example, RN50+Synth achieves 17.52% accuracy on ImageNet-D while RN50+Synth+DiffAug achieves 19.18% accuracy and can be further improved to 21.41%. Since these results are encouraging, we are curious to evaluate the performance of DiffAug against DiffAttack.
> ***
> **[Q8]** This is a good question! DE utilises a set of different diffusion time-steps ($\\mathcal{S}$ in Eq. 6) while DDS uses a single diffusion time-step. Crucially, DDS is based on the randomized smoothing theory in [1] whereas DE is inspired from test-time augmentation techniques [2]. Additionally, DE uses average voting for prediction whereas DDS uses majority voting. Further, the number of samples generated for each input is typically higher in DDS (e.g., 100 samples for ImageNet) as compared to DE (9 new samples when using $\\mathcal{S}=\\{0,50,100 … 450\\}$). In Figure 11, we compare the performance between DE and DiffAug: since DiffAug is equivalent to DDS using 1 sample, we expect DDS results to be similar to Fig. 11b.
>
> [1] Certified Adversarial Robustness via Randomized Smoothing
>
> [2] Understanding test-time augmentation.
>
> ***
> We hope this resolves all concerns and look forward to resolving any outstanding concerns during the discussion period.

---

> > ### Comment · Reviewer_QY34 · 2024-08-11
> > **Thanks for the response.**
> >
> > After carefully considering the authors' response, I have no further questions. I am pleased to recommend the acceptance of this paper, and I have raised my rating to 7.
> >
> > In the revised manuscript, I would be delighted to see not only the experiments on robustness against diffusion-based models as mentioned in Q6, but also those on the effect of DiffAug using conditional diffusion models, as discussed in Q3. Including these experiments would offer readers a more comprehensive understanding of the method's scope of application. This seems to be a point of curiosity for many, as also highlighted by Reviewer uEtP, and would make a valuable addition to the paper.

---

> > > ### Author Response · Authors · 2024-08-12
> > > **Thank you!**
> > >
> > > We thank you again for your insightful reviews and positive evaluation of our submission. We agree with your recommendation that DiffAug experiments with conditional diffusion models would be valuable. Based on both your and Reviewer uEtP’s interest in conditional diffusion models, and based on Reviewer 3vHc’s interest in Diffusion-Transformer models, we will include in the final version additional DiffAug experiments with Diffusion-Transformer (DiT). Since DiTs are trained for classifier-free guidance (i.e., both class-conditioning as well as null-conditioning), we will include additional settings with class-conditioning using DiTs in the final version.

---

### Official Review · Reviewer_3vHc · 2024-07-12

**Soundness:** 2
**Presentation:** 2
**Contribution:** 2
**Rating:** 6
**Confidence:** 4

**Summary:**

This paper applies a diffusion-based data augmentation method to enhance the robustness of classifier. First, a gaussian perturbation is applied to train examples and then a single diffusion denoising step is applied to generate the augmentations. Besides, DiffAug can also be combined with other augmentations to further improve robustness. Empirically, DiffAug can achieve improvements in classifier generalization, gradient quality and image generation performance.

**Strengths:**

1. DiffAug can be combined with other augmentations to improve robustness. Besides, DiffAug can be also used to improve many other performance, such as classifier generalization, gradient quality and image generation performance
2. DiffAug is simple, computational and easy to follow to improove robustness.

**Weaknesses:**

1. Absence of ablation study of sampling methods and sampling processes. Did the use of better sampling methods and more steps achieve better results?
2. Absence of ablation study of diffusion model. Can other diffusion model such as DiT also be used to apply DiffAug

**Questions:**

As shown in Weakness 1, why adopt a single diffusion denoising step when the current sampling method, such as DPM-Solver, can generate good results within 10 steps? In theory, better sample generation can achieve better results

**Limitations:**

yes

---

> ### Author Rebuttal · Authors · 2024-08-07
>
> We thank you for your detailed reviews. We thank you for appreciating the strengths of our method’s simplicity and computational efficiency and its ability to be combined with other augmentations to further improve robustness. In the following responses, we aim to address the weaknesses and answer your questions:
>
> > **_[Q1] Absence of ablation study of sampling methods and sampling processes. Did the use of better sampling methods and more steps achieve better results? Why adopt a single diffusion denoising step when the current sampling method, such as DPM-Solver, can generate good results within 10 steps? In theory, better sample generation can achieve better results._**
>
> This is a good question! To answer this question, we performed additional experiments during the rebuttal week using high-quality synthetic data from stable-diffusion as described in our global response (see [E1]). Specifically, we use 1.3M synthetic imagenet images in addition to the original ImageNet training data to finetune the torchvision resnet-50 model for 10 epochs. We call this RN50+Synth. To understand the utility of DiffAug in this case, we finetune another instance of the torchvision resnet-50 model for 10 epochs using DiffAug as well as additional synthetic data. We call this RN50+Synth+DiffAug.
>
> From our experimental evaluation across several datasets, we find that DiffAug training and DiffAug-Ensemble (DE) inference offers complementary benefits to extra synthetic training data. We agree with you that, theoretically, DiffAug augmentations generated using a DDPM model trained on ImageNet should not provide further improvements when high-quality synthetic data from SD — trained on LAION-5B, that also covers ImageNet — is already available. Yet, we surprisingly observe that DiffAug improves over and beyond additional high-quality synthetic data! To explain this, we first note that DiffAug is qualitatively different from previous diffusion-based augmentation techniques. Depending on the diffusion time used to generate the DiffAug augmentation, the resulting image can vary greatly in quality (as shown in Fig. 1). As a result, classifying some of these augmented images is more challenging as compared to the original examples producing a regularizing effect that leads to empirical robustness improvements. Lastly, yet importantly, the complexity introduced by DiffAug training does not sacrifice test accuracy despite training on poor quality examples. This is an unusual and noteworthy property that can be understood by interpreting denoised examples to lie on the image manifold following recent theoretical studies [E.g., refs. 9 & 34 in the main paper].
>
> In summary, DiffAug can enhance robustness even when efficient sampling techniques are available to synthesize high-quality images since its performance improvement can be attributed to the regularizing effect from learning to classify partially synthesized train examples (i.e.,  diffused-and-denoised examples). We will include these results in the final version of the paper and supplementary materials.
>
>
> ***
> > _**[Q2] Absence of ablation study of diffusion model. Can other diffusion model such as DiT also be used to apply DiffAug.**_
>
> While we consider the DDPM model (a variance-preserving (VP) SDE) for all our ImageNet experiments, we use a variance-exploding (VE) SDE diffusion in our classifier-guided generation experiments for the CIFAR10 dataset. We find that DiffAug introduces identical improvements for both DDPM and VE-SDE in terms of classifier generalization, perceptually aligned gradients, and improved image-generation performance. Yes, in theory, it should be possible to apply DiffAug with a diffusion-transformer model instead of the UNet architecture.
> ***
>
> We hope this resolves all concerns and look forward to resolving any outstanding concerns during the discussion period.

---

> > ### Comment · Reviewer_3vHc · 2024-08-10
> >
> > In fact, I believe the authors haven't addressed my concerns. For the first question, I was hoping to see attempts at different sampling methods. The authors proposed using high-quality synthetic data but neglected to explore sampling approaches. Regarding the second question, my main concern is with the network architecture of the diffusion model, not the training method (VPSDE or VESDE). I hope the authors can still address my concerns. Thank you.

---

> ### Author Response · Authors · 2024-08-12
> **Thank you for your response!**
>
> We thank the reviewer for promptly reading our rebuttal, stating their outstanding concerns, and giving us another opportunity to address their concerns. We truly appreciate this level of engagement.
>
> **Better Sampling Methods and More Steps.**
>
> We originally interpreted your question as follows: better images should yield better results, so wouldn’t we get “even better” results by using the “even better” generated images? To answer this, we considered a strong baseline: a large-scale high-quality synthetic dataset generated with a stronger diffusion model. In particular, we used high-quality synthetic data generated using Stable-Diffusion with 50 reverse-diffusion steps of the PNDM sampler. Then, we studied the role of DiffAug augmentations — generated with a single reverse-diffusion step (of DDIM, or equivalently DPM-Solver-1) — when extra synthetic images are already available. Our experimental results showed that, surprisingly, the single-step DiffAug augmentations introduce further improvements beyond extra synthetic images.
>
> We believe we now understand your question better and it seems that you were interested in applying DiffAug with more denoising iterations (on the diffused training image), or using a different solver (e.g. DPM-solver-2). We address both of these questions directly, followed by a discussion:
>
> **(1) More denoising steps**: In fact, we originally had the same initial intuition as the reviewer: that taking more denoising steps might yield better performance! In our preliminary analyses, we therefore did indeed explore a multi-step extension to DiffAug (e.g., DDIM). *Interestingly, we found that multiple de-noising steps did not help as much as samples from a single reverse-diffusion step*, which is why we did not pursue it further. This initially surprised us and prompted us to think about it more carefully — we provide a detailed discussion about this below in addition to our discussion in lines 162-167 of our original paper submission, where we briefly explain why we do not consider multi-step denoising techniques despite their potential to improve sample quality. However, we would be very glad to expand upon this explanation in the final version!
>
> **(2) Different sampling method**: Additionally, in the last few days, based on this reviewer’s suggestion, we tried using a single-step of reverse-diffusion of DPM-solver-2, both for training the classifier, and also as the sampling method at test-time for the DiffAug ensemble-technique (DE) that we describe in the paper. In particular, we first diffused the train example to a random diffusion-time $t$ and used a single reverse-diffusion step of the order-2 DPM solver to integrate the diffusion ODE backwards from diffusion time $t$ to $t=\epsilon$ with $\epsilon=0.001$ instead of $0$ for numerical stability. The results are shown in the Table below: *in both cases, while the resulting augmented images are of high visual quality, this did not work as well as the sampling strategy that we are already using* (again, discussion is below). This was very interesting for us to try out, and we feel that having tried this is clearly strengthening our paper and results, as it is consistent with our understanding so far of what DiffAug is doing.
>
> | Training Method |  |  |Evaluation  |
> |:---:|:---:|:---:|:---:|
> |  | Default     | DiffAug-Ensemble (DPM-Solver-1) | DiffAug-Ensemble (DPM-Solver-2) |
> | AM | 26.72 | 34.08 | 31.77$\downarrow$ |
> | AM+DiffAug/DPM-Solver-1 | 29.47 | 38.58 | 35.56$\downarrow$ |
> | AM+DiffAug/DPM-Solver-2 | 22.96 $\downarrow$ | 29.69$\downarrow$ | 25.16$\downarrow$ |
>
> (We use $\downarrow$ to denote lower performance due to the use of DPM-Solver-2 instead of the default DPM-Solver-1 used in our paper.)
>
> We now describe the benefits of augmentation with a single reverse-diffusion step in classifier-training (apart from computational efficiency) in the following:
>
> **i.  A single reverse-diffusion step generates examples on the image manifold that cannot be generated by multi-step samplers.** The generated sample lies in regions between high-quality samples and can be understood mathematically from Eq. 4, where we observe $\hat{\bf x}_t$ is a convex-sum over examples from the data distribution. Also, see Fig. 5 in the Appendix for an illustration on a toy dataset.
> In fact, surprisingly, these "low-visual quality" samples are valuable in a way that "high-visual-quality" samples are not! It turns out that we get the best results by including the full range of visual quality: in Appendix B.5.2, we include an ablation study to identify the comparative advantages of “_stronger_” DiffAug augmentations ($t \in [500,1000]$) and “_weaker_” DiffAug augmentations ($t \in [0,500]$). While stronger augmentations offer greater performance improvements on ImageNet-C and OOD detection, we find using the entire diffusion time-range tends to yield better performance across all evaluated tasks.

---

> > ### Author Response · Authors · 2024-08-12
> > **Thank you for your response! (contd.)**
> >
> > **ii. Improved samplers generate higher quality examples faster but may not offer regularization effects similar to DiffAug.** To demonstrate this, we conducted experiments using additional high-quality synthetic samples generated with Stable-Diffusion. Improved samplers such as DPM can help generate new synthetic data faster. While large-scale synthetic data can improve performance on ImageNet-R, ImageNet-Sketch and ImageNet-D, we show that both DiffAug training and also DiffAug-Ensemble (DE) inference offer additional improvements in each of these cases. We attribute the performance improvements from extra synthetic images to the manually designed prompts (Table 8 of [1]) and large-scale upstream dataset (LAION-5B) used to train stable-diffusion. On the other hand, the benefits of DiffAug training can be attributed to the regularization effect introduced by training over examples lying in regions between clean samples (and on the image manifold).
> >
> > **iii.** Informally, we also found that there is another subtle catch with increasing the number of iterations. Our current method requires essentially no hyper-parameter tuning: it just works.  However, we found that **DiffAug with multiple reverse-diffusion steps tended to generate augmented examples that effectively belong to a different class than that of the original training image.** This happens because we use the entire diffusion time range by default — in particular, images diffused farther could get augmented to an image that belongs to a different class. On one hand, we were able to address this, by, e.g. defining an upper limit $\tau$ such that the forward-diffusion step is applied with $t \le \tau$. However, any such solution that we considered ultimately introduced additional hyperparameters that required tuning, but without noticeable gains, and in fact generally missing out on the robustness improvements from stronger augmentations. Interestingly, while  DiffAug with one reverse-diffusion step also does not preserve the class-label for larger diffusion time $t$ and injects label-noise while training, the model can still "learn" from these images without underfitting since the label-noise is correlated with visual quality. For more details, please refer to lines 148-167 on page 4 of the main paper; our current discussion in the paper is brief, but we would be glad to expand it.
> >
> > For test time image adaptation, DDA uses multiple reverse-diffusion steps to deal with severe ImageNet-C corruptions and an improved sampler such as DPM can help accelerate sampling in this case. However, we note that DDA contains hyperparameters (in addition to the diffusion range $\tau$) that can strongly influence the test time performance. DiffAug-Ensemble is an alternative approach that is robust to hyperparameter choices (Appendix B.6) and demonstrates comparable or better improvements as compared to DDA using a single reverse-diffusion step. We feel that it may be advantageous to implement and evaluate a multi-step DiffAug-Ensemble technique using DPM-solver.
> > ***
> > In summary, the benefit (in terms of regularization and robustness) of our augmentations appears to be in where the augmented data points sit in relation to the data manifold, and the “effective” augmentations do not necessarily correlate with visual quality (as supported by our experiments including: synthetic data, multi-step denoising, DPM-solver-2 denoising, Appendix B.5.2 ablation). For this reason we find that using a single reverse-diffusion DDIM step is both crucial for improvements introduced by DiffAug training, and is robust in that it does not require any hyper-parameter tuning. We appreciate your insightful questions and recommendations, and we look forward to including appropriate parts of this discussion in the final version of the paper.
> >
> > For the final version, we will also implement DiffAug-Ensemble (for inference) using the DPM-solver and also include an analysis of hyperparameters (e.g., diffusion-range) and runtime. We will also include the results of our preliminary analysis where we generated DiffAug train augmentations using multiple steps (e.g., DDIM, DPM-Solver-2). We hope that these additional experiments can inspire multi-step extensions to DiffAug in future work. More specifically, while many multi-step samplers are impressively optimised for improved efficiency and sample quality (i.e., for humans), multi-step samplers for improved regularizations (i.e., for downstream neural models) likely require alternative designs. Finding these designs, and using them to build on DiffAug, would be a very exciting future research direction.
> >
> >
> > [1] Leaving Reality to Imagination: Robust Classification via Generated Datasets
> >
> > ***

---

> ### Author Response · Authors · 2024-08-12
> **Thank you for your response! (contd.)**
>
> **Augmentation with Diffusion-Transformers (DiTs):** In theory, DiffAug can be applied using DiTs. Since DiTs are latent-diffusion models, we note that the augmentations can be generated by directly applying the VAE decoder over the one-step denoised latent representation. Previous works (e.g., [1,2]) follow this technique to use external guidance functions — that operate on images/piano-rolls as input — to guide the latent-diffusion sampling. Anecdotally, our informal experiments have suggested that DiffAug works with a variety of models. While we are greatly interested to train/evaluate classifiers with DiffAug using DiT, we are unable to report results of this experiment during the rebuttal discussion week due to time and resource constraints. We have already started working on this and we will certainly include results with DiT in the final revision.
>
> [1] Universal Guidance for Diffusion Models. (ICLR 2024)
>
> [2] Symbolic Music Generation with Non-Differentiable Rule Guided Diffusion (ICML 2024)
> ***
> We hope these responses help address your concerns. If there are any other concerns, including other ablation analyses that you feel are important for inclusion in the final version, please let us know and we will gladly address your concerns/recommendations.
>
> ***

---

> > ### Comment · Reviewer_3vHc · 2024-08-12
> >
> > Thank you very much for the detailed reply, which has solved my main concern, I will update my score.

---

> > > ### Author Response · Authors · 2024-08-13
> > > **Thank you!**
> > >
> > > We thank you for carefully considering our response and engaging with us during the discussion period. We are confident that the experiments prompted by your reviews not only strengthen our contributions but also help provide a better understanding of our new augmentation technique.

---

### Official Review · Reviewer_gb3b · 2024-07-13

**Soundness:** 3
**Presentation:** 2
**Contribution:** 2
**Rating:** 6
**Confidence:** 4

**Summary:**

The paper explores the use of diffusion as a data augmentation technique to train robust classifiers. Specifically, it investigates whether a diffusion model trained without additional data can be leveraged to enhance classifier performance. The study shows utilizing a diffusion model trained without extra data to improve classifier robustness on Imagenet-C by 2 points, while train improved classifiers with just a single step of reverse diffusion.
The proposed method is evaluated on challenging benchmarks such as Imagenet-C and Imagine, utilizing Vision Transformer (ViT) and ResNet models.

**Strengths:**

1. Originality: The paper proposes an effective method of one-step diffusion augmentation for training robust image classifiers. While similar augmentation techniques using generative models have been explored previously, the utilization of a single-step diffusion process in this context is interesting. The idea is straightforward yet innovative, offering a novel approach to enhancing classifier robustness with minimal computational overhead.

2. Clarity: the paper is clearly written.

3. Quality: the paper includes ablation study and visualization. Say on diffusion time.

**Weaknesses:**

1. Soundness of results: the robustness is only evaluated on Imagenet-C (severity 5 only), which is only local corruption. Does the proposed method work when viewpoint, background, scale, style, texture change? What about the other severity?  Say can test on Imagenet-A [1], ImageNet-D [2], Imagenet-S [3], ImageNet-R [4]. Since it is diffusion-based augmentation, the reviewer wondered if it will help ImageNet-D the most since this one is diffusion-based testing. Evaluation on those will help understand the strengths and weakness of the proposed approach.

2. Not clear how useful is test-time augmentation, can add the ablation study.

3. Unclear why the proposed method can handle covariate shifts. Can the author explain intuitively?

The score will be updated based on whether evaluation concerns are addressed.

[1] https://openaccess.thecvf.com/content/CVPR2021/html/Hendrycks_Natural_Adversarial_Examples_CVPR_2021_paper.html
[2] https://openaccess.thecvf.com/content/CVPR2024/html/Zhang_ImageNet-D_Benchmarking_Neural_Network_Robustness_on_Diffusion_Synthetic_Object_CVPR_2024_paper.html
[3] https://arxiv.org/abs/1905.13549
[4] https://openaccess.thecvf.com/content/ICCV2021/html/Hendrycks_The_Many_Faces_of_Robustness_A_Critical_Analysis_of_Out-of-Distribution_ICCV_2021_paper.html

**Questions:**

How do the hyper parameters of the one-step diffusion set? If you add more noise, you may change globally, if only local noise, you can only handle local noise corruption.

Can the author show an ablation study on this tradeoff if there is any?

**Limitations:**

Yes.

---

> ### Author Rebuttal · Authors · 2024-08-07
>
> We thank you for the detailed review with suggestions to strengthen the contribution. We thank you for  recognizing the strengths of our method’s novelty, simplicity and computational efficiency. We are happy to see that you found our paper clearly written with nice visualisations. In the following responses, we aim to address the weaknesses and answer your questions.
>
> > ***Evaluation on other severity***
>
> Following this reviewer’s suggestion, we evaluate the models on all severity levels and include the results in the global PDF. We notice similar overall trends as those that we observed for severity=5. On average, we find that the DiffAug training and DiffAug-Ensemble inference leads to improved performances. We will include this in the final version of the paper.
>
> ***
> > **_Evaluation on Imagenet-A [1], ImageNet-D [2], Imagenet-S [3] and ImageNet-R [4]._**
>
> As per this reviewer’s suggestion, we carried out evaluations on ImageNet-A and ImageNet-D. For this evaluation, we also considered the additional classifiers we trained during the rebuttal week: in particular, we trained classifiers using synthetic datasets generated with stable-diffusion (please refer to the global response for more details) and denote this as RN50+Synth. We use RN50+Synth+DiffAug when trained using DiffAug in addition to the synthetic data. We note that ImageNet-R and ImageNet-S results are already included in the Appendix (Table 6) – we will label the paragraph describing these results on page 5 (line 220) to ensure that these results are easily accessible. We summarize our findings as follows:
> * DiffAug training introduces slight improvements in the default evaluation mode (i.e., directly evaluating on test examples) for these datasets while DiffAug-Ensemble (DE) inference introduces notable improvements for Imagenet-R, ImageNet-D and ImageNet-S.
> * On ImageNet-S, the average performance improves from 15.45 (using default evaluation) to 17.99 (using DE). For DiffAug-trained models, the average performance improves from 15.51 (default) to 18.26 (DE). In particular, DiffAug training and DE inference improves the RN50 performance from 7.12 to 12.52.
> * On ImageNet-R, we find similar observations as above. For example, DiffAug training and DE inference improves the accuracy of RN50 from 36.16 to 41.61. On average, DE enhances the performance of DiffAug-trained models from 44.63 to 46.35. The best performance on ImageNet-R is obtained when using additional synthetic data: RN50+Synth has an accuracy of 49.28. Using both DiffAug training and DE inference, we can improve the performance to 54.71.
> * On ImageNet-D, we observe that extra synthetic data from stable-diffusion helps enhance the robustness most as you had predicted. Interestingly, DiffAug helps to further improve robustness in this case. For example, RN50+Synth obtains an accuracy of 17.52 on ImageNet-D. Using DiffAug training in addition to extra synthetic data improves the performance to 19.18. When using DE-evaluation, we observe further improvements up to 21.41.
> * On ImageNet-A, we generally find that all models except ViT struggle on this task. In this case, we find that DE inference leads to a reduction in performance on this task (on average). DiffAug training neither improves nor negatively affects the model performance in this case although we find slight improvements in many cases.
>
> In summary, we find that DiffAug training improves classifier robustness in general and this can be further enhanced using DiffAug-Ensemble (DE) inference.
> ***
> > **_[W2] Not clear how useful is test-time augmentation, can add the ablation study._**
>
> [A] This is a good question, and we acknowledge that while we included this ablation in appendix B.6, the reviewers are not expected to look at appendix materials. We will gladly add a comment in the main body of the paper that highlights and refers the reader to these results.
>
> To summarize these ablation results here: we find in multiple experiments that the DiffAug-Ensemble (DE) inference technique does indeed improve robustness. DE has two hyperparameters: maximum diffusion time-step and step-size controlling the number of augmentations. Overall, we also find that DE is robust to hyperparameter choices.
> ***
> > _**[W3] Unclear why the proposed method can handle covariate shifts. Can the author explain intuitively?**_
>
> [A] We include a detailed answer in the global response providing an intuitive explanation of DiffAug's performance (in Q2). In summary, DiffAug generates augmentations of varying sample-qualities, each presenting a different level of challenge for the classifier. Intuitively, this produces a regularisation effect that helps improve various aspects of classifier robustness including covariate-shift. Notably, the complexity introduced by DiffAug does not lower the test accuracy despite the explicit use of low sample-quality images for classifier training.
> ***
> > _**[Q1] How do the hyper parameters [...] tradeoff if there is any?**_
>
> [A] This is a very good question, and the reviewer is correct that smaller diffusion times produce DiffAug examples with localised modifications whereas larger diffusion times produce global modifications. In all our experiments, we use the entire diffusion time range to generate the DiffAug examples. To understand the influence of diffusion time-range, we conducted an ablation study (see Appendix B.5.2) comparing between _weaker_ DiffAug augmentations ($t \in [0,500]$) and _stronger_ DiffAug augmentations ($t \in [500,1000]$). While stronger and weaker augmentations help enhance different aspects of classifier robustness — for example, stronger augmentations are more useful to enhance performance on ImageNet-C and OOD detection — using the entire diffusion time scale tends to work better across all tasks.
>
> ***
> We hope we have addressed all your concerns and look forward to discussing any outstanding concerns in the discussion period.

---

> > ### Comment · Reviewer_gb3b · 2024-08-07
> > **Thank you for your answer. My questions are answered.**
> >
> > Score updated accordingly.

---

> > > ### Author Response · Authors · 2024-08-07
> > > **Thank you!**
> > >
> > > We thank you for your prompt acknowledgement of our new results and insights. We feel that these new experiments, prompted by your review, indeed strengthen the contributions of the paper.

---

### Official Review · Reviewer_uEtP · 2024-07-15

**Soundness:** 3
**Presentation:** 3
**Contribution:** 2
**Rating:** 5
**Confidence:** 3

**Summary:**

This paper proposes a data augmentation method DiffAug for training robust classifiers. The method is very simple: first, add Gaussian noise to one training image, and then denoise the noisy images with a pre-trained diffusion model. They also propose methods for test-time augmentation using DiffAug. Experiments show that DiffAug is able to improve robust accuracy on a wide range of datasets such as ImageNet-C, R,S, OOD detection, and certification accuracy. This paper also shows that the proposed method can improve the generation quality of classifier-guided diffusion generation.

**Strengths:**

1. To the best of my knowledge, this method is new. Most of the existing methods use diffusion models to generate new training examples instead of augmenting existing ones. The proposed method is simple and easy to follow and implement.

2. This paper provides extensive ablation experiments to demonstrate the effectiveness of the proposed method. The proposed method is able to achieve significant improvements even combined with SoTA augmentation for robustness, say AugMix and DeepAugment, which indicates the proposed method provides new information for the model to learn. DiffAug can also be used to improve the performance of guidance classifiers in diffusion generation, thus lead to better results.

3. Models trained with this method have perceptually aligned gradients and this paper provides nice figures to demonstrate this.

**Weaknesses:**

1. **Missing related work and comparison**

Comparison with existing work that uses large synthetic data datasets with diffusion models to improve training is lacking. For example [1] uses diffusion models to generate extra training examples to improve empirical adversarial robustness and [2] for certificated robustness. Both two works show significant improvements for robustness.

Specifically, suppose DiffAug train the model with E epochs on a dataset of N images. SoTA diffusion models can generate new images with K steps (K can be small (K<= 10) using DDIM, even 1 step using some flow based methods). If we use the diffusion model to generate EN/K new images, and train the model on the original dataset with the generated images. The extra computational cost is the same as DiffAug, if I understand it correctly. Will DiffAug achieve better results over this method? If yes, what could be the motivation/theory?


2. **Concerns regarding some numbers reported**

In the DeepAugment paper, they report results for ResNet50 on ImageNet-C ([3], Figure 5). The accuracy of DeepAugment+AugMix (DAM in this paper) is close to 60%, which is much higher than the numbers in this paper (around 40%). Similarly, the results of AugMix reported in the original paper are also much higher than the numbers in this paper. (I do not work in the corruption robustness field, so there may be errors in citing these numbers)

Can the authors explain why the gap is so large? If it is due to different settings. I suggest following the original settings for a fair comparison.

Similarly, the results on Table 8 are also significantly higher than numbers in [4], and far from SoTA.





[1] Better Diffusion Models Further Improve Adversarial Training. ICML23

[2] A Recipe for Improved Certifiable Robustness. ICLR 2024

[3] The Many Faces of Robustness: A Critical Analysis of Out-of-Distribution Generalization

[4] (CERTIFIED!!) ADVERSARIAL ROBUSTNESS FOR FREE!

**Questions:**

1. I do not pretty understand the sentence in line 128: what does "additional optimization objective" refer to?

```
When combining DiffAug with such novel augmentation techniques, we simply include Eq. 5 as an additional optimization objective instead of stacking augmentations
```

and line 131: why stacking augmentations are not good and what is the difference between stacking augmentations the the proposed method?
```
our preliminary analysis on stacking augmentations showed limited gains over simply training  the network to classify independently augmented samples likely because training on independent  augmentations implicitly generalizes to stacked augmentations.
```

2. Why use unconditional diffusion models? Will using conditional diffusion models raise to some problem?

3. Can authors provide some insights why the proposed method can improve both corruption robustness  and certificated adversarial robustness? Will this method also improve empirical adversarial robustness?

**Limitations:**

Yes, authors adequately addressed the limitations.

---

> ### Author Rebuttal · Authors · 2024-08-07
>
> We thank you for the insightful review with suggestions to strengthen the contribution. We thank you for your appreciation of our method’s novelty, simplicity and computational efficiency, our presentation/figures and extensive ablation experiments. In the following responses, we aim to address the weaknesses and answer your questions.
>
> > _**[W1-A] Comparison [...] is lacking.**_
>
> We address your suggestion for a comparison in the experiment [E1] described in the global response and include the results in Table 1 of the global PDF. In summary, we find that DiffAug training and DiffAug-Ensemble (DE) inference offer performance improvements _over and beyond_ additional high-quality synthetic datasets.
> ***
> > _**[W1-B] Missing Related Work.**_
>
> We thank you for bringing these papers to our attention, and we will certainly include discussion in the final version. We identify important distinctions and similarities between our work and the suggested papers. While the mentioned works focus exclusively on adversarial robustness, we focus on regular non-adversarial classifier training with an aim to improve different aspects of robustness (e.g., covariate shifts, OOD detection and certified adversarial accuracy). Nevertheless, both example papers demonstrate use of synthetic data generated with diffusion models trained with no extra data to improve the adversarial training technique and this is in fact very close to one of our research objectives: exploring the augmentation potential of diffusion models trained with no extra data. On the contrary, previous generative data augmentation approaches for regular non-adversarial classifier training often rely upon synthetic datasets from large diffusion models such as Stable-Diffusion or Imagen.
>
> ***
>
> > **_[W1-C] Specifically, [...] theory?_**
>
> Thank you for this interesting question about performance and sample quality vs compute budget! We provide a detailed answers to this in the global response (please see answers to Q1 and Q2). In the context of adversarial training, we imagine that corresponding enhancements in performance are likely possible using DiffAug. Additional exploration on adapting DiffAug to adversarial training and empirical validation is out of scope, but is an interesting avenue for future work.
>
> ***
>
> **[W2]** We appreciate the reviewer’s dedication to provide a well-informed review of this submission! We use official checkpoints and confirm our reproduction is error-free; we will release both the source-code and model checkpoints for reproducibility.
>
> The DeepAugment+AugMix result in the original paper was obtained by considering all 5 severities of ImageNet-C, while Table 1 is for severity=5 (please see Fig. 1 in the global PDF for results over entire ImageNet-C). Our reproduction of the ImageNet-test and ImageNet-R accuracies of these models do also match the official results. Since the ImageNet-C dataset contains 750k test samples for each severity level, considering all 5 severities is computationally expensive for some methods (e.g., DDA) and in those cases, the standard practice is to evaluate on severity=5 since this represents the most challenging examples.
>
> As we have described in lines 259-267 and as correctly identified by this reviewer, DDS already achieves state-of-the-art certified robustness results by using a pretrained 305M-parameter BeIT-L network. It is intuitive that training a classifier with DiffAug enhances the DDS certified accuracy and we empirically confirm this in Table 8 for a 86.6M parameter ViT-B model since this is computationally feasible.
>
> ***
> **[Q1]** Additional optimization objective refers to extra loss term. When extending previous methods with DiffAug, we modify their official code by simply adding $\mathcal{L}$ in Eq. 5 to the original loss-terms. For example, AugMix involves 2 loss-terms: a cross-entropy loss term and a Jensen-Shannon divergence loss term. When combining it with DiffAug, we simply introduce Eq.5 as the third loss-term.
>
> Stacking refers to the sequential application of distinct augmentations on the same image. While stacking augmentations may intuitively offer more robustness enhancements, our preliminary analysis showed no advantage of stacking augmentations over simply including the DiffAug loss in addition to the original training loss. Therefore, we choose to implement DiffAug as an additional loss term since this requires minimal change to the original code and fewer design choices (e.g., order of augmentations).
> ***
>
> **[Q2]** Similar to many standard augmentation techniques, we do not rely upon labels for DiffAug broadening its applicability -- e.g., DiffAug can also be applied at test-time when labels are unknown. In theory, DiffAug with conditional diffusion models should not cause any problem. However, this introduces additional hyperparameters such as the guidance-strength and choice of prompt. Additionally, it requires two forward passes through the diffusion model (one unconditional and one conditional) per training step. If the guidance-strength is very high, the classifier could cheat by exploiting imperceptible image statistics (e.g., due to adaptive layer-norms). This is an interesting extension for future exploration.
>
> ***
> **[Q3]** In summary, DiffAug generates augmentations of varying sample-qualities, each presenting a different level of challenge for the classifier. Intuitively, this produces a regularisation effect that helps improve various aspects of classifier robustness including covariate-shift. Since both DiffAug and DDS utilise a single reverse-diffusion step, we can observe improvements in certified adversarial accuracy. Yes, we can also improve empirical adversarial robustness against $l_2$ perturbations by using a majority vote following the Algorithm 2 in [1].
>
> [1] (CERTIFIED!!) ADVERSARIAL ROBUSTNESS FOR FREE!
> ***
> We hope we have addressed all your concerns and look forward to discussing any outstanding concerns in the discussion period.

---

> > ### Author Response · Authors · 2024-08-12
> > **Additional Information regarding Q2**
> >
> > We thank you again for your thorough review of our submission and insightful questions. Based on both Reviewer QY34’s recent positive recommendation and your question (Q2) about conditional augmentation, we want to inform you that we will include experiments on conditional DiffAug in the final version. In particular, we will explore class-conditioning with Diffusion-Transformer (DiTs) models, as that is also consistent with the DiT experiments we committed to include in the final version to Reviewer 3vHc. We hope we have addressed all your concerns and look forward to discussing any outstanding concerns in the remaining discussion period.

---

> > > ### Comment · Reviewer_uEtP · 2024-08-12
> > >
> > > Thank authors for the response, and my concerns are addressed. I will keep my score as my original score is based on that the numbers are reported correctly and the proposed method should be better than using  generated data to increase the dataset size.

---

> > > > ### Author Response · Authors · 2024-08-13
> > > > **Thank you!**
> > > >
> > > > We thank you again for your thorough reviews and insightful suggestions for additional experiments to further strengthen our contributions.
> > > >
> > > > We also thank you for carefully considering our response, acknowledging that your concerns are addressed and maintaining your original score in support of acceptance.

---

### Author Rebuttal · Authors · 2024-08-07

We thank all the reviewers for their insightful and thoughtful reviews with suggestions for improvements. We are happy to see that the reviewers appreciated the novelty (uEtP, gb3b) and simplicity/computational efficiency (uEtP, gb3b, 3vHc, QY34) of the method and found the paper to be well-presented (uEtP, gb3b, QY34) with extensive experiments and analysis (uEtP, QY34).

Based on the reviews, we conducted additional experiments during the rebuttal week and include the results in the attached PDF. We describe the experimental details as follows:

**[E1] Experiments with additional high-quality synthetic data**

For this experiment, we consider 1.3 M synthetic images --- released by [1] --- generated with Stable-Diffusion (SD) using diverse prompts aimed at enhancing classifier robustness. We fine-tune the torchvision resnet-50 model for 10 epochs using both real and synthetic images and call this RN50+Synth. Next, we repeat this finetuning process with DiffAug and call this RN50+Synth+DiffAug.  We evaluate these models across many datasets and include the results in ***Table 1*** of the global PDF.

From Table 1, we immediately observe that large-scale synthetic data can improve performance on ImageNet-R, ImageNet-Sketch and ImageNet-D. However, in each of these cases, we can observe that both DiffAug training and also DiffAug-Ensemble (DE) inference each offer additional improvements. For example, DiffAug training and DE inference can improve performance on ImageNet-R from 49.28 (RN50+Synth, Default (Def)) to 54.71 (RN50+Synth+DiffAug, DE). Similarly, DiffAug training and DE inference improves performance on ImageNet-Sketch (35.45 to 37.39) and ImageNet-D (17.52 to 21.41).

On ImageNet-C, we observe that extra synthetic data does not offer any improvement while DiffAug training and DE inference offer similar improvements both with and without extra synthetic data.
***
**[E2] More Datasets**

We present further empirical evidence in favour of DiffAug-Training and DE inference in ***Table 2*** (ImageNet-A/D) and ***Figure 1*** (ImageNet-C, all severities).

***

Now, we address some common questions:

**Q1. [uEtP, 3vHc] DiffAug vs Additional high-quality data generated with more steps.**

This is an interesting question! We answer this question based on the experiment [E1] by comparing between RN50+Synth and RN50+Synth+DiffAug. Based on our experimental results in Table 1, we interestingly observe that DiffAug using improved-DDPM offers improvements over and beyond additional synthetic data using Stable-Diffusion. This is surprising due to the following reasons:
1. Stable-Diffusion is trained on LAION-5B, a much larger dataset that also subsumes ImageNet while the improved-DDPM model is trained on ImageNet data alone.
2. Additional synthetic data requires more compute per each sample (e.g., 50 reverse-diffusion steps) whereas DiffAug just uses one reverse-diffusion step.

Even when using a diffusion model trained on a smaller dataset, DiffAug manages to offer performance improvements _complementary_ to high-quality synthetic examples. We imagine that a better diffusion model for generating DiffAug augmentations may provide further performance improvements. Ultimately, when using comparable diffusion models (e.g., same training data and similar parameter counts) for synthetic data augmentation as well as DiffAug, +Synth+DiffAug may likely be equivalent to +DiffAug allowing for a compute-efficient diffusion-based augmentation technique that combines the advantages of both extra training data as well as DiffAug. We leave this exploration for future work.

***

**Q2. [uEtP, gb3b] Intuitive Explanation of DiffAug's Performance**

We do have some intuition, which we present here, as to why DiffAug offers additional improvements, even beyond what we obtain by making direct use of 1.3M high-quality synthetic images.

As a first step (and to state what might be obvious), the improvements in ImageNet-R, ImageNet-Sketch and Imagenet-D when fine-tuned with extra synthetic images can be attributed to the large-scale upstream dataset (i.e., LAION-5B) used to train Stable-Diffusion and various prompts designed to enhance data-diversity (e.g., Table 8 of [1]). To intuitively understand improvements from DiffAug training, we first note that DiffAug augmentation is qualitatively different from previous diffusion-based augmentation methods: depending on the diffusion time used to generate the DiffAug augmentation, the resulting image can vary greatly in quality (as shown in Fig. 1). As a result, classifying some of these augmented images is more challenging as compared to the original examples producing a regularizing effect that leads to empirical robustness improvements. Lastly, yet importantly, the complexity introduced by DiffAug training does not sacrifice test accuracy despite training on poor quality examples. This is an unusual and noteworthy property. We can explain this observation by interpreting denoised examples to lie on the image manifold following recent theoretical studies [E.g., refs. 9 & 34 in the main paper].
We can use this interpretation of denoised examples to explain why DiffAug-Ensemble enhances robustness: a diffuse-and-denoise transformation applied to a test example from a different distribution *projects it towards the manifold of the original distribution*. This data manifold intuition helps clarify how and why these examples are fundamentally different from examples augmented with pure gaussian noise.
***

[1] Bansal and Grover. (2023) Leaving Reality to Imagination: Robust Classification via Generated Datasets

---

### Decision · Program_Chairs · 2024-09-25

**Decision:**

Accept (poster)

**Comment:**

This work demonstrates that data augmentation with a single diffusion step improves robustness to input corruptions and adversarial attack and it likewise improves OOD detection with small but consistent gains, and furthermore demonstrates the improvement compounds with other techniques.

Four reviewers are mostly in favor of acceptance (QY34: 7, gb3b: 6, 3vHc: 6) except there is one vote for borderline rejection (uEtP: 5). The authors provide a thorough rebuttal by answering questions and supplying additional results as requested (such as addtional benchmarks like ImageNet-A/D/R and further evidence of complementary effects vs. purely training on synthetic data). These scores follow the rebuttal by the authors, where three of four reviewers raised their scores while uEtP acknowledged the rebuttal and maintained their sore of 5. In more detail, uEtP acknowledged the rebuttal and maintained their score now that concerns are addressed and the results are as they seem ("the numbers are reported correctly and the proposed method should be better than using generated data to increase the dataset size").

The AC sides with acceptance in agreement with the majority of the reviews. The authors are encouraged to address the discussion of unconditional and conditional diffusion models (QY34, uEtP) and follow through on the rebuttal suggestion to include DiTs with class conditioning.